# NEUROSURF: NEURAL UNCERTAINTY-AWARE ROBUST SURFACE RECONSTRUCTION

## ABSTRACT

Neural implicit functions have become popular for representing surfaces because they offer an adaptive resolution and support arbitrary topologies. While previous works rely on ground truth point clouds, they often ignore the effect of input quality and sampling methods on the reconstruction. In this paper, we introduce NeuroSURF, which generates significantly improved qualitative and quantitative reconstructions driven by a novel sampling and interpolation technique. We show that employing a sampling technique that considers the geometric characteristics of inputs can enhance the training process. To this end, we introduce a strategy that efficiently computes differentiable geometric features, namely, mean curvatures, to augment the sampling phase during the training period. Moreover, we augment the neural implicit surface representation with uncertainty, which offers insights into the occupancy and reliability of the output signed distance value, thereby expanding representation capabilities into open surfaces. Finally, we demonstrate that NeuroSURF leads to state-of-the-art reconstructions on both synthetic and real-world data.

## 1 INTRODUCTION

It is important to reconstruct surfaces for downstream tasks. Finding a suitable surface representation is also crucial. Many methods have been proposed to train a neural network to reconstruct the surface using point cloud or mesh data. Moreover, most of the methods consider only dense and clean input. How to deal with sparse and noise input is undiscussed. Some methods require (oriented) surface normals to produce satisfactory results. However, in real-world scenarios, the requirement on (high quality) point cloud and its (oriented) normals is often hard to fulfill, which leads to failure cases of recent reconstruction methods Gropp et al. (2020); Chibane et al. (2020b). Moreover, training data might be unevenly distributed. The selection of sampling strategies impacts both the efficacy of training and the quality of the outcomes. A natural question arises: can we adjust the sampling method according to the features of the input data and enhance the training? To address these questions, in this paper, we propose a technique that utilizes depth images as basic input and predicts the signed distance value of query points. Depth images enable us to easily estimate point normals and other surface geometry features such as curvatures. We propose a curvature-guided sampling strategy, which enhances the training process and reduces the uneven sampling problem. Moreover, this method effectively interpolates among inputs to counter the low-quality data issues. We also incorporate an uncertainty value in neural implicit representation to ascertain the reliability of the prediction. This augments the capabilities of the signed distance field to depict open surfaces without additional effort, as shown in Fig. 1. In summary, our contributions are the following.

- We introduce NeuroSURF, which can deal with sparse input data to reconstruct surfaces ranging from single objects to large scenes using the same framework.
- We propose a method that computes mean curvature directly from input depth images and uses it as a curvature-guided sampling strategy, which considers the geometric feature of the input to enhance the training efficiency.
- We introduce an uncertainty-aware implicit neural representation that gives an uncertainty of predictions and enables open surface presentation.
- Extensive experimental studies show that our proposed method achieves state-of-the-art reconstruction results on challenging synthetic and real-world datasets.

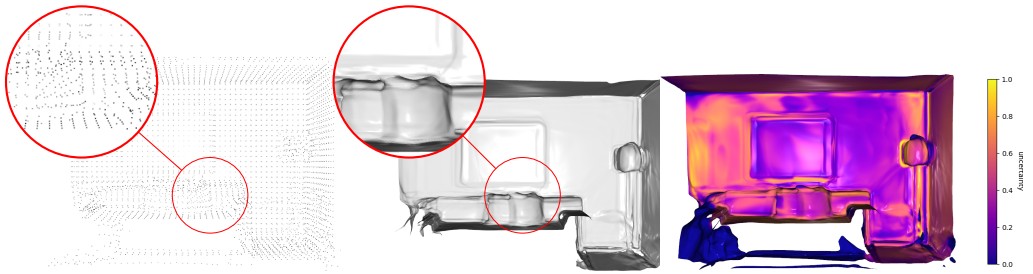

Figure 1: Reconstructed the surface (middle) of the proposed method with only a sparse input (left, $\sim$ 6k points) together with the uncertainty of each surface point (right).

## 2 RELATED WORK

### 2.1 SURFACE REPRESENTATIONS AND RECONSTRUCTIONS

**Surface representation**    Surface representation can be classified into two categories based on the stored surface properties: explicit surface representation (e.g., polygon mesh or point clouds) and implicit surface representation (e.g., signed distance field). Explicit methods struggle with complex topologies, resolution adjustments, local modifications, and potential high memory consumption when storing high-resolution surfaces. In contrast, implicit representations represent the surfaces by storing indirect information about the surface, which overcomes the shortages of explicit methods. However, **classical** implicit methods still suffer from fixed resolution and high memory consumption issues. For example, the memory consumption is $\mathcal{O}(n^3)$ for a $n$ resolution voxel grid. Luckily, **neural** implicit representations can encode the surface implicit information such as occupancies Mescheder et al. (2018); Genova et al. (2019); Chibane et al. (2020a), signed/unsigned distance functions Park et al. (2019); Gropp et al. (2020); Novello et al. (2022); Chen & Zhang (2019); Michalkiewicz et al. (2019); Chibane et al. (2020b) into neural networks. One can query any 3D points in space to obtain the corresponding attributes. This approach allows for recovering highly detailed surfaces at a lower memory cost than traditional surface representation methods such as classical signed distance field (SDF). Moreover, it is a continuous representation suitable for further mathematical analysis of the represented surface. (Signed/unsigned) Distance Field stores the shortest (signed) distance of a given point to a surface, The gradients of an SDF provide a normal vector of the surface, but using SDF requires well-defined "inside" and "outside" notions, which restrict representing open surfaces. An unsigned distance field (UDF) does not distinguish the inside or outside. However, the lack of a sign introduces ambiguity in surface reconstruction, such as the correct surface orientation.

**Surface Reconstructions**    Much like with surface representations, surface reconstructions techniques can be categorized into traditional approaches, including **classical** methods like Poisson surface reconstructions (Kazhdan et al., 2006) and SSD (Calakli & Taubin, 2011). These techniques commonly utilize point clouds as input to create a polygon mesh. Unfortunately, these traditional methods are considerably reliant on the quality of input data, often failing when faced with complex geometries. Moreover, using traditional methods is hard to modify output resolution without repeating the complete reconstruction process. On the other hand, **learning-based** implicit surface reconstruction approaches train neural networks to generate surface features Chibane et al. (2020a); Gropp et al. (2020); Sitzmann et al. (2020); Michalkiewicz et al. (2019); Park et al. (2019), allowing for changing the output resolution without retraining the network. Additionally, they can handle relatively complicated geometry. The typical input of these methods is 3D data, e.g., a point cloud or a mesh. Some methods also need a point cloud with normals information to ensure a satisfactory result Sitzmann et al. (2020); Gropp et al. (2020). However, obtaining point clouds with normals is not straightforward if the ground-truth mesh is not given or the point cloud quality is poor, which often happens in real-world applications.

### 2.2 TRAINING DATA SAMPLING

**Sparse training data**    Learning-based methods sample data from the given input to train the network to predict underlying surface features. Sampling the training data is easy when the input format, such as a mesh Novello et al. (2022), allows for infinite sampling. When only a limited sample is available, such as in point clouds Sitzmann et al. (2020); Gropp et al. (2020), random sampling from the point cloud is the most common way. It leads to one problem: inefficient training data leads to bad reconstruction results Chibane et al. (2020b;a).

**Biased Sampling** Another problem of the random sampling is that points in the point cloud, especially the points acquired from real-world data, are not uniformly located on the surface, as discussed by Yang et al. (2021) and Novello et al. (2022). Points extracted from the iso-surface will likely be gathered near high-curvature areas Yang et al. (2021). Plus, complex surface areas need more points to represent their features. Random sampling does not consider these effects. To avoid this issue, (Yang et al., 2021) samples on and near the iso-surface with some tolerance. Novello et al. (2022) proposes to sample according to principal curvatures of the surface points such that the sampled points are evenly distributed according to the curvatures. They divide points into low, medium, and high curvature categories according to the absolute sum of two principal curvature $\kappa_1$ and $\kappa_2$. However, ground-truth meshes are required in their computation pipeline.

To tackle the challenges mentioned above, in this paper, we focus on more practical inputs: depth images, which can be directly acquired from hardware. We initiate a coarse voxel grid that retains initial SDF values, normals, and curvatures derived from depth images, effectively addressing the issues related to input points with (oriented) normals. The voxel grid structure allows us to locate any query points in space to get the related attributes by efficiently interpolating within voxels, thereby resolving the sparse input problem. Our approach leverages a hybrid representation combining point clouds and voxels, improving implicit surface reconstruction accuracy. Moreover, we embed an uncertainty value into implicit surface representation to indicate the reliability of the SDF value. The uncertainty value helps to eliminate redundant areas and facilitates the reconstruction of open surfaces. The paper is structured as follows: we introduce our method in Sec. 3, the experiments and evaluation results are stated in Sec. 4, we summarize our method in Sec. 5, more results and analysis are in Appendix A.

## 3 METHOD

Our goal is to train a network $f(\mathbf{x}, \theta) : \mathbb{R}^3 \to \mathbb{R} \times [0, 1]$, which predicts the SDF value and uncertainty of this value, such that the wanted surface $\mathcal{S}$ lies on the level-set $\{\mathbf{x} | f(\mathbf{x}) = 0\}$. We assume the shape $\mathcal{S}$ is captured by a set of depth images $\{D_k\}$. We first utilize a coarse voxel grid from depth images. During the process, points normals and curvatures are computed and merged to the coarse voxels. Then we introduce a curvature-guided sampling method together with a novel interpolating method to create and select training data for each epoch. Last, we talk about extracting surface considering uncertainty. We also show that our method can easily be transplanted to other methods to improve the results.

### 3.1 VOXEL UTILIZATION

We utilize a coarse voxel grid $\{\mathbf{v}_i\} \subset \mathbb{R}^3$ for $i \in \mathcal{V}$ containing SDF value for each voxel following the method described in Sommer et al. (2022). For each voxel $\mathbf{v}_i \in \mathbb{R}^3$, we initialize two local properties: the SDF value $\psi_i^v \in \mathbb{R}$ of that voxel and the uncertainty of the SDF value $w_i^v \in [0, 1]$. Larger $w_i^v$ means the SDF value is reliable, and $w_i^v = 0$ indicates the voxel properties are not updated. Different from the traditional voxel grid, a gradient of distance $\mathbf{g}_i^v \in \mathbb{R}^3$ is also integrated for each voxel after computing the normal vector of each pixel in the depth images using FALS method (Badino et al., 2011). The point cloud contained in the voxel can be extracted by

$$\mathbf{x}_i = \mathbf{v}_i - \hat{\mathbf{g}}_i^v \psi_i^v \tag{1}$$

where $\hat{\mathbf{g}}_i^v = \frac{\mathbf{g}_i^v}{\|\mathbf{g}_i^v\|}$. For details, please refer to Sommer et al. (2022) and the Appendix A

**Curvature integration** Depth images can provide more geometric information other than normals Kurita (1999); Di Martino et al. (2014). To solve the biased sampling problem mentioned in Sec. 2.2, we propose directly incorporating mean curvature during the voxelization step to help with the sampling procedure during training. The mean curvature and other differential geometry features, such as the Gaussian curvature, are closely related to principal curvatures, often denoted as $\kappa_1$ and $\kappa_2$. The mean curvature $H$ is the average of the principal curvatures, while the Gaussian curvature $K$ is their product. They are both local geometry properties of the surface and reveal the local topology characteristics. In this way, we do not need a ground-truth mesh for computing curvature information. To our knowledge, we are the first to integrate mean curvatures, computed from depth images, with voxels for efficient sampling afterward. A depth image can be viewed as a Monge patch of a surface, i.e. $z = D(m, n), (m, n) \in \Omega \subset \mathbb{R}^2$ with pixel coordinates $(m, n)$ lay

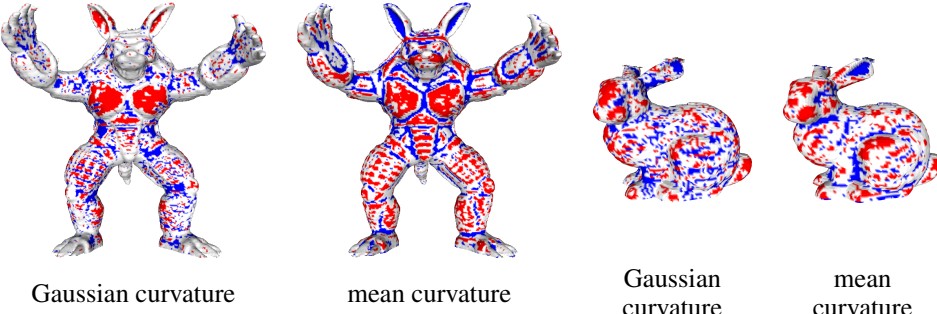

Gaussian curvature    mean curvature    Gaussian curvature    mean curvature

Figure 2: The visualization of Gaussian curvatures and mean curvatures of each point. The red color indicates a high curvature area, and the blue color indicates a low curvature area. A positive mean curvature ($H > 0$) signifies a convex surface, while a negative mean curvature ($H < 0$) indicates a concave surface. Positive Gaussian curvature ($K > 0$) indicates that the surface is locally shaped like a dome or sphere at that point, and negative Gaussian curvature ($K < 0$) indicates that the surface is locally saddle-shaped or hyperbolic at the point.

in the image domain $\Omega$ (do Carmo, 1976; Spivak, 1999). Thus, the Monge patch $\mathcal{M} : \Omega \to \mathbb{R}^3$ is $\mathcal{M}(m, n) = (m, n, D(m, n))$. To compute the two types of curvatures from the depth

$$K(m,n) = \frac{D_{mm}D_{nn} - D_{mn}^2}{(1 + D_m + D_n)^2} , \tag{2}$$

$$H(m,n) = \frac{(1 + D_m^2)D_{nn} - 2D_mD_nD_{mn} + (1 + D_n^2)D_{mm}}{2(1 + D_m^2 + D_n^2)^{3/2}} , \tag{3}$$

where $D_m = \frac{\partial D(m,n)}{\partial m}$ is the partial derivative of depth w.r.t. $x$-axis in the image plane; similarly, $D_n = \frac{\partial D(m,n)}{\partial n}, D_{mn} = \frac{\partial^2 D(m,n)}{\partial m \partial n}$ for $D_{mm}$ and $D_{nn}$. The computation is done on-the-fly per depth image. After associating a curvature to each point from every depth image, we update voxel curvature attributes by averaging all points inside the same voxel $K_i^v = \frac{1}{N} \sum_j K(\mathcal{M}(m,n)_j)$, where $i$ is the voxel index and $j$ is the pixel index. Curvatures are computed using local image coordinates, and we can still fuse curvatures into voxel (world) coordinates because the mean curvature $H(m,n)$ and Gaussian curvature $K(m,n)$ are invariant to changes of the parameterization on the smooth surface represented by $\mathcal{M}(m,n)$ do Carmo (1976). A detailed explanation is provided in the Appendix A. The normal and curvature computation time for a $480 \times 640$ depth image, plus updating voxel attributes according to this incoming depth, is around $50$ millisecond. Fig. 2 displays that the computed curvature indeed captures the local geometric properties of the surface.

## 3.2 VOXEL-BASED SAMPLING

**Interpolating Sampling**  In this section, we introduce an interpolation strategy that deals with the sparse input and uses the gradient and curvature information. With the initialized coarse voxel representation $\{\mathbf{v}_i\}$, we first sample a random point $\mathbf{p} \in \Gamma$ in 3D space, where $\Gamma \subset \mathbb{R}^3$ is the sampling domain. With the help of the voxel grid structure, we can localize in which voxel the point $\mathbf{p} \in \mathbb{R}^3$ lies and denote the coordinate of the voxel center as $\mathbf{v}(\mathbf{p}) \in \mathbb{R}^3$. Then, the signed distance value of the

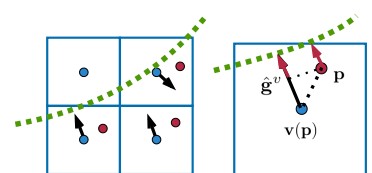

Figure 3: Illustration of equation 4

sampled point $\mathbf{p}$ can be easily computed by Taylor expansion with the help of the stored gradient $\mathbf{g}^v$ as Fig. 3

$$\psi^p = \psi^v + \langle \hat{\mathbf{g}}^v, \mathbf{p} - \mathbf{v}(\mathbf{p}) \rangle , \tag{4}$$

where $\hat{\mathbf{g}}^v = \frac{\mathbf{g}^v}{\|\mathbf{g}^v\|}$ is the normalized gradient of the distance field. The uncertainty of the sampled point is interpolated using

$$w^p = \frac{v_s - \psi^x}{v_s} w^v , \tag{5}$$

where $v_s$ is the voxel size, $w^v$ is initial voxel uncertainty as mentioned in Sec. 3.1. The equation 5 sets the maximum value to the points on the surface ($\psi^p = 0$) and reduces the value when points are

moved away from the surface. Meanwhile, the point $\mathbf{p}$ inherits the curvatures of its voxel $K^v$ and $H^v$. Thus, we are not restricted to sampling only from a fixed set of points but can pick any point in the space for training. Furthermore, locating $\mathbf{v}(\mathbf{p})$ can be done by one-step calculation, see Appendix A. Our method does not require any additional nearest neighbor search or access to neighbor voxels. The equation 4 is within its own voxel. Hence it is free from the voxel resolution since the training inputs are points-signed distance value pairs.

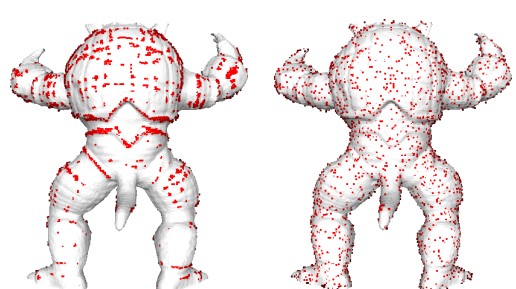

**Curvature-guided sampling** To avoid the uneven sampling problem, we divide sampled points into low, median, and high curvature regions, similar to Novello et al. (2022), we use the mean (or Gaussian) curvature instead of principle curvatures. Due to the page constraint and the similarity of the mean and Gaussian curvatures (see Fig. 2), in the following, we only show the results based on the mean curvature. For each epoch, $m$ points (see Sec. 3.2) are sampled from low curvature category $\hat{\mathbf{p}} \sim \{\mathbf{p} \in \Gamma \mid H^p < \underline{H}\}$, median curvature category $\hat{\mathbf{p}} \sim \{\mathbf{p} \in \Gamma \mid \underline{H} \le H^p < \bar{H}\}$, and high curvature category $\hat{\mathbf{p}} \sim \{\mathbf{p} \in \Gamma \mid H^p \ge \bar{H}\}$. The threshold $\underline{H}$ and $\bar{H}$ are chosen by the curvature range of $0.3$ and $0.7$ percentile. The visualized sampling results are shown in Fig. 4.

Figure 4: Points gathering on high curvature effect (left) and our sampling results after considering points mean curvature (right).

### 3.3 UNCERTAINTY-AWARE IMPLICIT SURFACE RECONSTRUCTION

**Loss with uncertainty** We would like to recover the neural implicit function $f : \mathbb{R}^3 \to (\psi, w) \subset \mathbb{R} \times [0, 1]$, such that the surface lies on the level-set $\{\mathbf{x} \mid f(\mathbf{x}) \in 0 \times (\tau, 1]\}$, and $w$ is the uncertainty of the predicted signed distance value $\psi$. $\tau$ is the uncertainty threshold. Given a point $\mathbf{p}$ in the sample domain $\Gamma$, as described in Sec. 3.2, its corresponding voxel $\mathbf{v}(\mathbf{p})$ with the interpolated SDF value $\psi^p$ and uncertainty $w^p$ (using equation 4 and equation 5), we define the loss function of the geometric and the normal constraints as

$$l_{\mathcal{X}}(\theta) = \frac{1}{|\Gamma^+|} \int_{\Gamma^+} (|\psi - \psi^p|) \mathrm{d}\Gamma \,, \tag{6}$$

$$l_{\mathcal{N}}(\theta) = \frac{1}{|\Gamma^+|} \int_{\Gamma^+} (1 - \langle \frac{\nabla_\psi f(\mathbf{p}, \theta)}{\|\nabla_\psi f(\mathbf{p}, \theta)\|}, \hat{\mathbf{g}} \rangle) \mathrm{d}\Gamma \,, \tag{7}$$

$$l_{\mathcal{W}}(\theta_r) = \int_{\Gamma} |w - w^p| \mathrm{d}\Gamma \,, \tag{8}$$

where $\Gamma^+$ indicates the area with the sampled uncertainty $w^p > 0$. The equation 7 evaluates the cosine similarity of surface normal and implicit function gradient. The final loss is

$$l(\theta, \theta_r) = l_{\mathcal{X}}(\theta) + \tau_n l_{\mathcal{N}}(\theta) + \tau_w l_{\mathcal{W}}(\theta_r) + \tau_e l_{\mathcal{E}}(\theta) \,. \tag{9}$$

**Surface extraction with uncertainty** It is reasonable to consider uncertainty when extracting surface using the Marching cubes algorithm Lorensen & Cline (1987), as the uncertainty indicates the reliability of the signed distance value. Moreover, we also need to deal with the voxel grid with $w = 0$. As shown in Fig. 5, A single zero uncertainty vertex leads to a line (instead of a triangle), while two of this kind lead to a point Botsch et al. (2010); Farin (2002). Therefore, we can naturally reconstruct open surfaces, thanks to our uncertainty estimation.

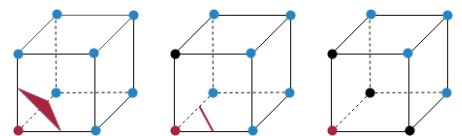

Figure 5: surface extraction with uncertainty. Black vertices represent zero uncertainty points. Red and blue vertex mean points with negative and positive SDF values, respectively.

### 3.4 INCORPORATING WITH OTHER METHODS

Our curvature-guided sampling can seamlessly incorporate popular implicit surface reconstruction methods such as Gropp et al. (2020); Sitzmann et al. (2020). These methods take point clouds as

input. The points with curvature can be extracted using equation 1 and the geometric loss equation 6 is

$$l_{\mathcal{X}}(\theta) = \frac{1}{|\mathcal{V}|} \sum_{i \in \mathcal{V}} \|f(\mathbf{x}_i; \theta)\|_1 \; . \tag{10}$$

Additionally, our interpolating method simplifies the method that needs nearest neighbor search, such as Neural-Pull (Ma et al., 2020). Neural-Pull proposes to train a network to learn the pulling direction and the distance to the iso-surface. During training, the method samples random points $\mathbf{p}$ and uses nearest neighbor search to find the closest surface point $\mathbf{x}$. Then, a network $f$ is trained to minimize the loss

$$l_{\mathcal{X}}(\theta) = \frac{1}{|\Gamma|} \int_{\Gamma} |\mathbf{x} - \frac{\nabla f(\mathbf{p})}{\|\nabla f(\mathbf{p})\|} f(\mathbf{p})| \mathrm{d}\Gamma , \tag{11}$$

with $\tau_n = 0$, $\tau_e = 0$. Our interpolating method ( Sec. 3.2) eliminates the nearest neighbor search required in the original Neural-Pull. As for a random point $\mathbf{p}$, after finding the corresponding voxel $\mathbf{v}(\mathbf{P})$, the closest surface point $\mathbf{x}$ can be easily located using equation 1. In Sec. 4, we show that it effectively reduces the noise during training and leads to better reconstruction quality. The modified Neural-Pull outperformed the original implementation, especially under sparse inputs.

Even though it is based on voxel representation, the proposed sampling method can access surface points without a surface extraction step such as Marching cubes. During the interpolating step, there is no nearest neighbor search or cubic interpolation, which needs to access 8 voxel vertices like in most voxel-based methods. The only interpolation step is done within its voxel by equation 4. We use this hybrid representation to lead an efficient sampling step.

## 4 EVALUATION

To demonstrate that our sampling method improves the robustness and accuracy of the implicit surface learning. We validate our method on synthetic and real-world datasets, including objects and scene scenarios. For synthetic datasets with ground-truth mesh, we rendered perfect depth images and camera poses using the ground-truth mesh to initialize voxel grid $\{\mathbf{v}_i\}$. To test low-quality input, we initialed two different voxel resolutions, $64^3$ (**sparse**) and $256^3$ (**dense**), to vary the sparsity of the input. The real-world datasets contain RGB-D sequences with noisy, sparse depth images and noise camera poses. To compare with the method that takes a point cloud as input, the point cloud is extracted from voxels using equation 1 with normals. We use an 8-layer multi-layer perceptron (MLP) with ReLU activations. Each layer has 256 nodes, and the last layer has 2 output nodes for the SDF and uncertainty. We set the learning rate to $10^{-4}$ with decay and use the mean curvature for curvature-guided sampling. The batch size is $10k$, and we train for $10k$ epochs for each dataset. Our PyTorch implementation takes approximately 15 minutes to train on a GeForce GTX TITAN X GPU with CUDA for each dataset. All meshes are extracted with the Marching cube algorithm with grid resolution $128^3$.

### 4.1 VOXEL-BASED INTERPOLATING SAMPLING

To verify that interpolating sampling helps when the input data is noisy and sparse, we compare against the classical methods, SSD Calakli & Taubin (2011) and Poisson surface reconstruction Kazhdan et al. (2006), as well as learning-based methods, IGR Gropp et al. (2020) and SIREN Sitzmann et al. (2020), which take a point cloud **with normals** as the input. IF-NET Chibane et al. (2020a) also uses a voxelized point cloud as input. We test our method on $4$ synthetic datasets to get the quantitative error. After extracting the mesh, we computed the Chamfer distance (CD) and Hausdorff distance (HD) of the reconstructed mesh with respect to the ground-truth mesh in two different input (sparse and dense) resolutions. Fig. 6 shows the visual comparison and the Tab. 1 shows the quantitative comparison. All methods produce satisfactory results when the input points are dense. However, our method still gives satisfactory results when the input is sparse. NDF Chibane et al. (2020b) trains a network to learn UDF of a surface to represent an open surface. However, the output is only a denser point cloud, and we failed to recover the mesh using Ball-Pivoting Algorithm as the author described in the paper. Additionally, we show that our interpolating method improves the previous method. The Neural-Pull method (Ma et al., 2020) outperforms the methods use equation 10 as the geometric loss Gropp et al. (2020); Sitzmann et al. (2020), when the normal information is not available. However, nearest neighbor search fails easily when the points are too sparse or noisy, while

| Metric | Dataset | | Method | | | | | | | |
|---|---|---|---|---|---|---|---|---|---|---|
| | | | SSD | Poisson | IF-NET | SIREN | IGR | IGR (curv) | OURS | OURS (curv) |
| CD ($\times10^2$) | Bunny | sparse | 0.204 | 0.278 | 0.303 | 8.745 | 0.481 | 0.447 | 0.068 | **0.067** |
| | | dense | 0.224 | 0.242 | 0.315 | 7.582 | 0.501 | 0.491 | 0.073 | **0.068** |
| | Armadillo | sparse | 0.038 | 0.031 | **0.025** | 2.396 | 0.060 | 0.034 | 0.039 | 0.037 |
| | | dense | **0.024** | 0.025 | 0.026 | 2.331 | 0.035 | 0.034 | 0.031 | 0.030 |
| | Dragon | sparse | 0.210 | 0.206 | 0.158 | 2.710 | 0.209 | 0.198 | 0.179 | **0.157** |
| | | dense | 0.151 | 0.150 | 0.181 | 2.784 | 0.162 | 0.154 | 0.130 | **0.126** |
| | Happy Buddha | sparse | 0.180 | 0.240 | 0.258 | 2.717 | 0.307 | 0.259 | 0.135 | **0.125** |
| | | dense | 0.210 | **0.202** | 0.258 | 2.720 | 0.248 | 0.240 | 0.214 | 0.267 |
| HD | Bunny | sparse | 0.057 | 0.074 | 0.112 | 0.817 | 0.149 | 0.142 | 0.026 | **0.016** |
| | | dense | 0.071 | 0.065 | 0.107 | 0.816 | 0.153 | 0.155 | 0.020 | **0.012** |
| | Armadillo | sparse | 0.014 | 0.006 | 0.007 | 0.357 | 0.018 | 0.008 | 0.015 | **0.006** |
| | | dense | 0.005 | 0.006 | **0.004** | 0.301 | 0.014 | 0.013 | 0.007 | 0.007 |
| | Dragon | sparse | 0.043 | 0.050 | 0.037 | 0.340 | 0.037 | 0.039 | **0.030** | 0.040 |
| | | dense | 0.040 | 0.039 | 0.036 | 0.317 | 0.056 | 0.045 | 0.030 | **0.025** |
| | Happy Buddha | sparse | 0.048 | 0.061 | 0.059 | 0.338 | 0.092 | 0.081 | 0.023 | **0.017** |
| | | dense | 0.075 | 0.057 | 0.058 | 0.332 | 0.059 | 0.057 | 0.054 | **0.044** |

Table 1: Error numbers for our method with (last column) and without curvature sampling (last second column) and comparison methods. It shows that our method achieves better accuracy under sparse and dense inputs.

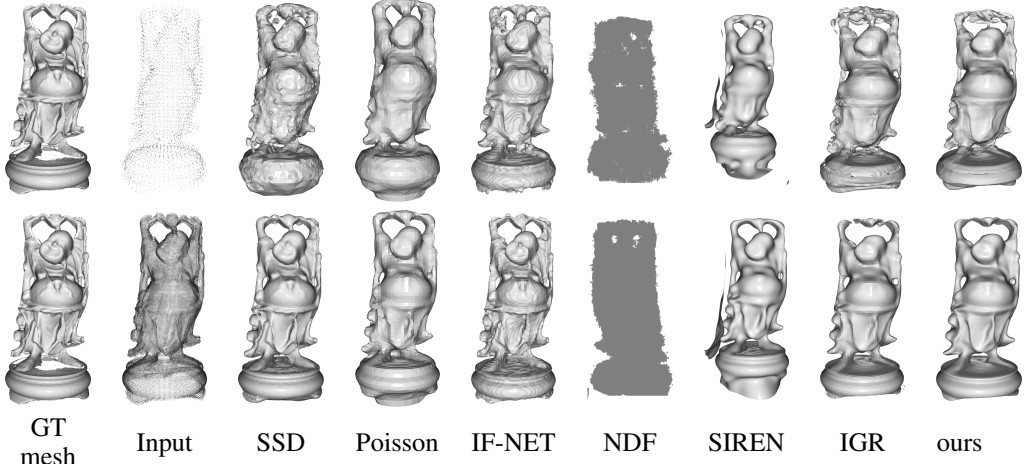

| GT mesh | Input | SSD | Poisson | IF-NET | NDF | SIREN | IGR | ours |

Figure 6: Comparison results with IGR Gropp et al. (2020), SIREN Sitzmann et al. (2020) and IF-NET Chibane et al. (2020a) with two different density input on synthetic datasets. Sparse input *happy_buddha* Curless & Levoy (1996) has $\sim 5k$ points, and dense one has $\sim 96k$ points.

our interpolating sampling strategy overcomes the problem of sparsity and noise. In Fig. 7, we show the performance of our sampling method with loss equation 11 compared to the original Neural-Pull method and the baseline method IGR. Our method and Neural-Pull do not use normals in their loss terms, while IGR still uses point normals. The first two rows are synthetic datasets, and the last two rows are real-world datasets *vase* and *sokrates* Zollhöfer et al. (2015). These two datasets are RGB-D sequences filming a real object. Each dataset contains around 40 depth images. The noise originates from both the depth images and camera poses. IGR Gropp et al. (2020) works well for sparse and dense inputs on synthetic datasets, whereas Neural-Pull fails in sparse situations. Our method compares favorably across all scenarios.

## 4.2 RANDOM SAMPLING VS. CURVATURE-GUIDED SAMPLING

We show that curvature-guided sampling helps in two aspects. First, it stabilizes the learning procedure, leading to a faster convergence of the minimum solution. We show this visually by rendering reconstructed meshes during early training epochs to see the learning efficiency of the different sampling methods. We compute these rendered meshes' Chamfer distance (CD) and Hausdorff distance (HD) to show that curvature-guided sampling has a smoother error curve. Fig. 9 shows that the network converges faster with curvature sampling than with a random sample. In Fig. 8, the solid lines and dash lines are Chamfer distance errors of curvature-guided sampling and random sampling, respectively. We normalized the CD errors by dividing the maximum error within each dataset to draw all lines in one figure. The curvature-guided sampling lines have a smoother trend and reach a lower error faster.

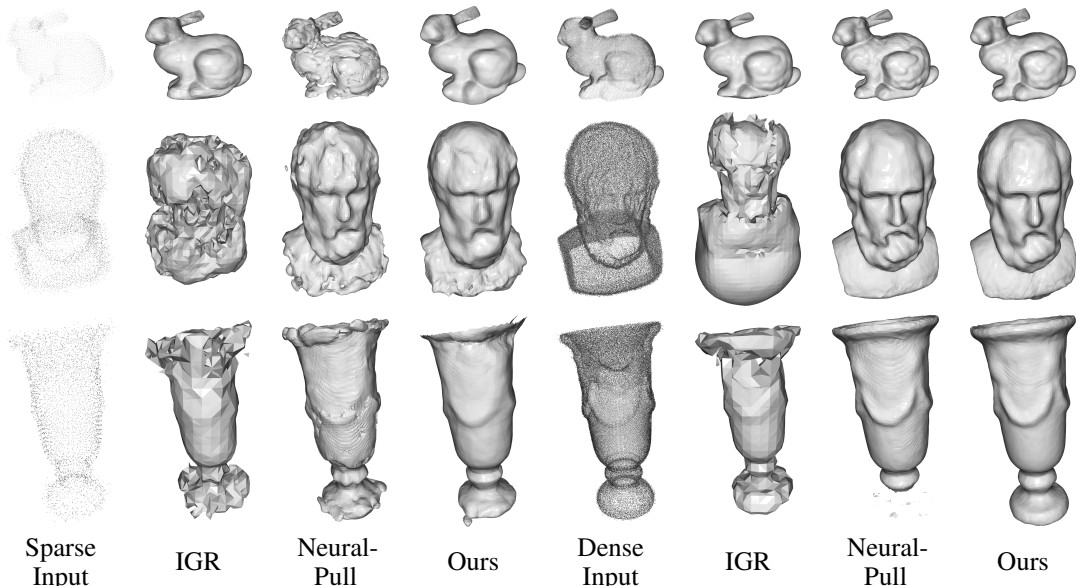

Figure 7: Comparison results with NeuralPull Ma et al. (2020) and IGR Gropp et al. (2020) with two different density input. Sparse input on synthetic dataset *bunny* has $\sim 5k$ points and dense one with $\sim 100k$ points. Two real-world datasets *sokrates* have $\sim 9k$ points and $150k$ points in sparse and dense situations, *vase* has $\sim 4k$ and $\sim 81k$ in two situations. The figure shows the results are largely improved by only changing to our sampling strategies.

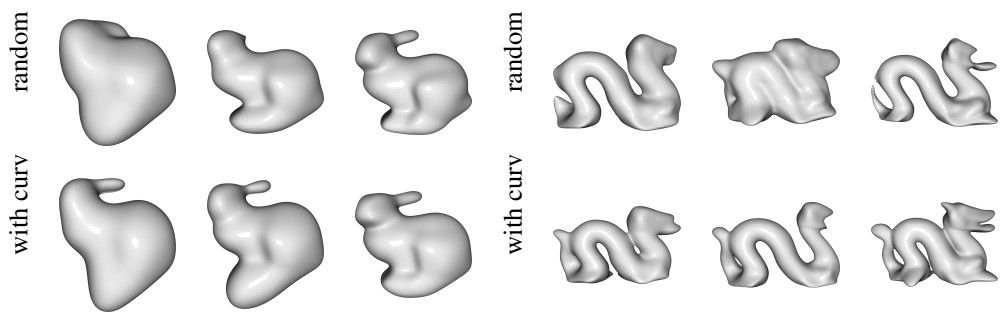

Figure 9: We extract surface during training to compare the effect of curvature-guided sampling and random sampling.

Second, we show that curvature-guided sampling also increases the accuracy of reconstructed meshes. We test our method with and without curvature-guided sampling (last two columns in Tab. 1) for a comparison. Moreover, we test on IGR Gropp et al. (2020) and change its sampling method during training from randomly choosing points to curvature-guided sampling as described in Sec. 3.2(last third and fourth columns in Tab. 1). Both comparison pairs show that considering curvature information during training improves the results.

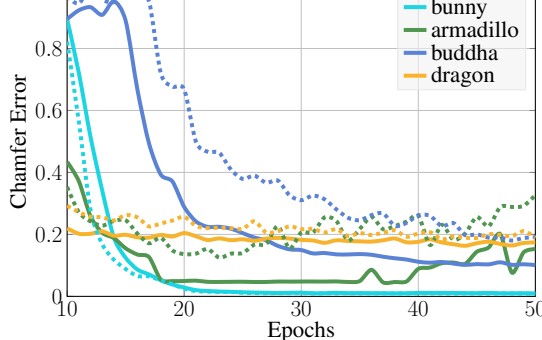

Figure 8: The CD error during different training epochs.

### 4.3 UNCERTAINTY PREDICTION

In this section, we show the uncertainty prediction result, which illustrates that the uncertainty helps to eliminate redundant areas. We focus on showing the results on the scene dataset to show that with the help of uncertainty, we can also represent

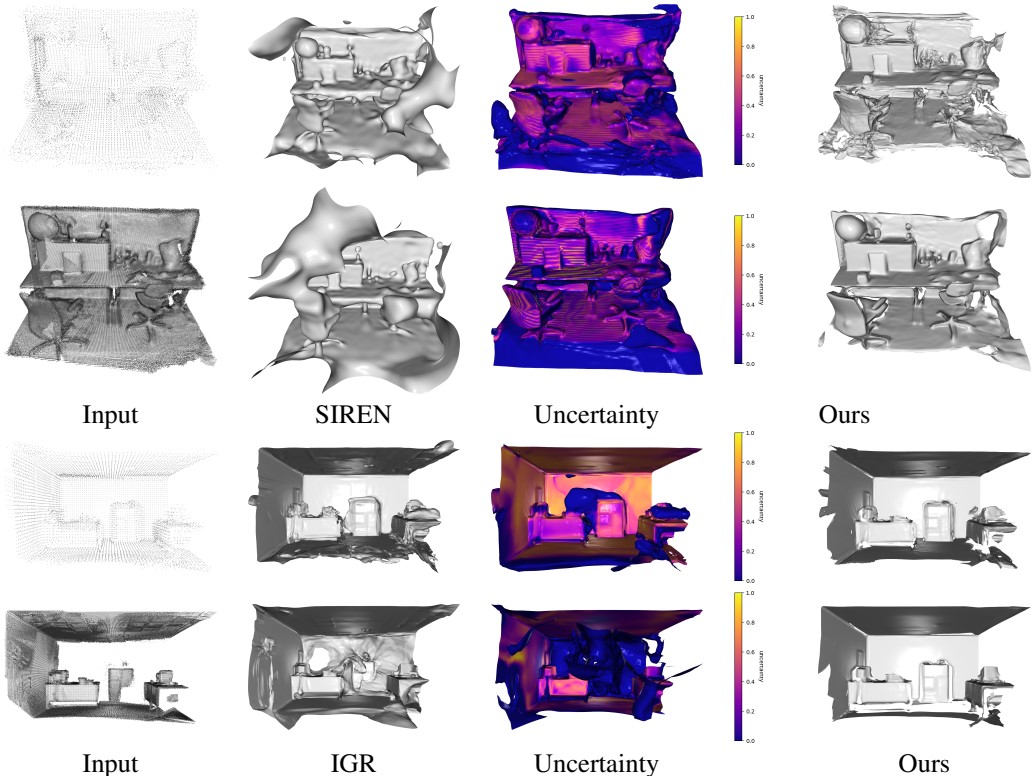

Figure 10: Scene reconstruction results with different density input on real-world dataset *TUM_rgbd* (first two rows, sparse points $\sim 14k$, dense points $\sim 330k$) with noisy camera poses as well. Synthetic dataset *icl_nium* (last two rows, sparse points $\sim 14k$ and dense points $\sim 215k$) with ground truth camera poses.

an open surface. Many previous works Sitzmann et al. (2020); Chibane et al. (2020b) have also trained neural networks to represent scene-level surfaces. However, a method such as Sitzmann et al. (2020) produces extra artifacts outside the surface. Although the authors propose one term in the loss function to penalize off-surface points for creating SDF values close to 0, it can not eliminate all artifacts, especially when the input is sparse and noisy. This problem can be solved by considering uncertainty during surface extraction as described in Sec. 3.3. Due to space constraints, we show a subset of comparison results. For more results, please refer to Appendix A.

## 5 CONCLUSION

**Summary**    In this work, we have presented a novel surface reconstruction method which integrated with a sampling method that can efficiently deal with low-quality inputs. Our approach operates on depth images, which can be directly acquired from hardware. We propose a method that computes surface geometric properties: normals and curvatures on depth images instead of using ground truth mesh. We integrate normals and curvatures into a coarse voxel grid to enable interpolating in sparse point clouds. The proposed method efficiently deals with sparse and noisy input and improves the reconstructed quality. Moreover, the curvature-guided sampling and interpolating method can be easily incorporated into different implicit surface reconstruction methods. The by-product uncertainty gives a reliability indication for the predicated signed distance value and can help with non-closed surface representations.

**Limitations**    Our approach does not specialize in surface completion and, therefore, fails to recover any missing areas in the input point cloud or depth data, which implies its inability to handle occlusion.

**Future Work**    The techniques presented in this work can be easily integrated with other neural reconstruction methods and shape completion techniques. Moving forward, our plan is to incorporate shape completion and neural rendering techniques to deal with missing areas.

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

# A APPENDIX

## A.1 CODE, DATASETS AND BASELINE METHODS

Our code and evaluation scripts will be publicly available upon acceptance. We will also provide the detailed information about the code of baseline methods.

| | name | type | year | link | license |
|---|---|---|---|---|---|
| Choi et al. (2016) | Redwood | dataset | 2016 | `http://www.redwood-data.org/3dscan/` | Public Domain |
| Curless & Levoy (1996) | The Stanford 3D | dataset | 1994 | `http://graphics.stanford.edu/data/3Dscanrep/` | Public Domain |
| Zollhöfer et al. (2015) | multi-view dataset | dataset | 2015 | `http://graphics.stanford.edu/projects/vsfs/` | CC BY-NC-SA 4.0 |
| Handa et al. (2014) | ICL-NUIM | dataset | 2014 | `https://www.doc.ic.ac.uk/~ahanda/VaFRIC/iclnuim.html` | CC BY 3.0 |
| Sturm et al. (2012) | TUM-rgbd | dataset | 2012 | `https://cvg.cit.tum.de/data/datasets/rgbd-dataset` | CC BY 4.0 |
| Sommer et al. (2022) | gradient-SDF | code | 2022 | `https://github.com/c-sommer/gradient-sdf` | BSD-3 |
| Gropp et al. (2020) | IGR | code | 2020 | `https://github.com/amosgropp/IGR` | - |
| Sitzmann et al. (2020) | SIREN | code | 2019 | `https://github.com/vsitzmann/siren` | MIT license |
| Chibane et al. (2020a) | IF-NET | code | 2020 | `https://virtualhumans.mpi-inf.mpg.de/ifnets/` | - |
| Ma et al. (2020) | Neural-Pull | code | 2021 | `https://github.com/bearprin/neuralpull-pytorch` | - |
| Chibane et al. (2020b) | NDF | code | 2020 | `https://virtualhumans.mpi-inf.mpg.de/ndf/` | - |
| Kazhdan et al. (2006) | Poisson | code | 2006 | `http://www.open3d.org/` | - |
| Calakli & Taubin (2011) | SSD | code | 2011 | `http://mesh.brown.edu/ssd/software.html` | - |

Table 3: Used datasets and code in our submission, together with reference, link, and license. We did our real-world experiments on two datasets, multi-view dataset Zollhöfer et al. (2015) (for which ground truth poses exist), and Redwood Choi et al. (2016) (without ground truth poses). Two synthetic dataset, the Stanford 3D Curless & Levoy (1996), which is an object dataset, and ICL-NUIM dataset Handa et al. (2014), which is a scene dataset. For the comparison methods, we use the code listed in the table.

## A.2 MATHEMATICAL DETAIL

### A.2.1 MATH NOTATIONS

We summarize important math notation we used in the paper and appendix in Table 4.

| Symbol | Description | Symbol | Description |
|---|---|---|---|
| $\mathbf{x} \in \mathbb{R}^3$ | 3D points | $\mathcal{P} \subset \mathbb{R}^3$ | point cloud set |
| $\mathcal{V} \subset \mathbb{N}^+$ | points index set | $\mathcal{S} \subset \mathbb{R}^3$ | continuous surface |
| $f(\mathbf{x}, \theta)$ | neural implicit function | $\theta \in \mathbb{R}^{n \times m}$ | learnable parameter |
| $w^p \in [0, 1]$ | points uncertainty | $w^v \in [0, 1]$ | voxel uncertainty |
| $\psi^v \in \mathbb{R}$ | voxel SDF value | $\psi^p \in \mathbb{R}$ | point SDF value |
| $\hat{\mathbf{g}}^v \in \mathbb{R}^3$ | normalized distance gradient | $\mathbf{g}^v \in \mathbb{R}^3$ | voxel distance gradient |
| $\nabla$ | differential operator | $H^p \in \mathbb{R}$ | point mean curvature |
| $\Gamma \subset \mathbb{R}^3$ | sample domain | $\gamma \in \mathbb{R}$ | SDF threshold |
| $K \in \mathbb{R}$ | Gaussian curvature | $H \in \mathbb{R}$ | mean curvature |
| $k_1, k_2$ | principal curvature | $D \subset \mathbb{R}^2$ | depth image |
| $\Omega \subset \mathbb{R}^2$ | image domain | $\mathcal{M} \subset \mathbb{R}^3$ | Monge path |
| $d_{\mathcal{S}}(\cdot)$ | signed distance to surface $\mathcal{S}$ | $\Gamma^+ \subset \mathbb{R}^3$ | sample domain with positive weight |
| $\bar{H} \in \mathbb{R}$ | higher threshold | $\underline{H} \in \mathbb{R}$ | lower threshold of curvature |
| $\mathbf{n}_i \in \mathbb{R}^3$ | known points normal | $\boldsymbol{Q} \in \mathbb{R}^{3 \times 3}$ | camera intrinsic matrix |
| $\boldsymbol{R} \in SO(3)$ | camera rotation matrix | $\mathbf{t} \in \mathbb{R}^3$ | camera translation vector |

Table 4: Summary of our notation in the main paper and the supplementary material.

### A.2.2 Voxelization Details

Given an incoming depth $D(m, n), (m, n) \in \Omega$ with $z = D(m, n) \in \mathbb{R}$ and the estimated pose $\boldsymbol{R}, \mathbf{t}$, the 3D points in world coordinates are

$$\mathbf{x} = \boldsymbol{R}\boldsymbol{Q}^{-1} \begin{bmatrix} m \\ n \\ 1 \end{bmatrix} z , \tag{12}$$

$$\boldsymbol{Q} = \begin{bmatrix} f_x & 0 & c_x \\ 0 & f_y & c_y \\ 0 & 0 & 1 \end{bmatrix} , \tag{13}$$

where $Q$ is the camera intrinsic matrix. The SDF value of points $\mathbf{x}_j$ and its normal $\mathbf{n}_j$ and curvature $K_j$ computed from the depth image $D$, then the voxel grid $\{\mathbf{v}_i\}$ SDF and weight is computed by

$$d_{\mathcal{S}}(\mathbf{v}_i) = (\mathbf{x}_{j^*} - \mathbf{v}_i)^\top \hat{\mathbf{g}}_i \tag{14}$$

$$\nabla d_{\mathcal{S}}(\mathbf{v}_i) = \boldsymbol{R}\mathbf{n}_{j^*} \tag{15}$$

$$w^v(\mathbf{v}_i) = \begin{cases} 1, & d_{\mathcal{S}}(\mathbf{v}_i) > 0 \\ 1 + \frac{d_{\mathcal{S}}(\mathbf{v}_i)}{v_s T}, & d_{\mathcal{S}}(\mathbf{v}_i) > -v_s T \\ 0, & \text{else} \end{cases} \tag{16}$$

$$j^* = \arg\min_j \|\mathbf{x}_j - \mathbf{v}_i\| \tag{17}$$

where $v_s$ is the voxel size, $T \in \mathbb{N}^+$ is the truncate voxel number, in this paper, voxel size is set to $0.8cm$ for $64^3$ grid and $0.2cm$ for $256^3$ grid with $T = 5$. Iterating over all depth image, the SDF $\psi_i^v$, weight $w_i^v$, gradient $\mathbf{g}_i$ and curvature $H_i^v$ is updated by

$$\psi_i^v \longleftarrow \frac{w_i^v \psi_i^v + w^v(\mathbf{v}_i) d_{\mathcal{S}}(\mathbf{v}_i)}{w_i^v + w^v(\mathbf{v}_i)} \tag{18}$$

$$\mathbf{g}_i \longleftarrow \frac{w_i^v \mathbf{g}_i^v + w^v(\mathbf{v}_i) \boldsymbol{R}\mathbf{g}_i^v}{w_i^v + w^v(\mathbf{v}_i)} \tag{19}$$

$$K_i^v \longleftarrow \frac{w_i^v K_i^v + w^v(\mathbf{v}_i) K_{j^*}}{w_i^v + w^v(\mathbf{v}_i)} \tag{20}$$

$$w_i^v \longleftarrow w_i^v + w^v(\mathbf{v}_i) \tag{21}$$

### A.2.3 Proof of the Curvature Integration

In the paper section 3.1, we mention that the transformation between two depth coordinates has non-zero Jacobian (non-zero determinant); hence, integrating curvatures from depth images makes sense. Here is the formulation and proof.

*The determinant of the Jacobian of the parameter transformation of the parameterization in the two depth images is non-zero; the mean curvature $H(x, y)$ and Gaussian curvature $K(x, y)$ are invariant.*

*Proof.* Given two depth images $D_1$ and $D_2$ taken at two different positions. Suppose the transformation from position 1 to position 2 is a rigid body motion $\boldsymbol{T} = [\boldsymbol{R}, \mathbf{t}]$, where $\boldsymbol{R} \in SO(3)$ is a rotation matrix and $\mathbf{t} \in \mathbb{R}^3$ is a translation vector. Let pixel $p = (m, n)$ in $D_1$, $0 \neq z = D_1(m, n)$, and $\boldsymbol{Q} \in \mathbb{R}^{3\times3}$ be the camera intrinsic matrix, then the transformation of pixel $p$ to $\bar{p} = (\bar{x}, \bar{y})$ in $D_2$ is

$$\bar{z} \begin{bmatrix} \bar{x} \\ \bar{y} \\ 1 \end{bmatrix} = \boldsymbol{Q}\mathbf{x} \,, \tag{22}$$

where $\mathbf{x}$ is computed using equation 12 and $\boldsymbol{Q}$ is same as equation 13. is invertible, $f_x$, $f_y$ is the camera focal length and $c_x$, $c_y$ is the principal points. The pixel $3D$ coordinates in under two camera view is $\boldsymbol{Q}^{-1}(x, y, 1)^\top z$ and $\boldsymbol{Q}^{-1}(\bar{x}, \bar{y}, 1)^\top \bar{z}$. Let $\{r_{ij}\}_{ij}, i, j \in \{1, 2, 3\}$ be the element in $\boldsymbol{R}$ and $\mathbf{t} = (t_1, t_2, t_3)^\top$, compute the right side, we have $\bar{z} = r_{31}x + r_{32}y + r_{33}z + t_3$ is the depth value after a rigid body motion and $\bar{z} \neq 0$ since it does not fall to image plane of $D_2$ as we assume the point is visible in both camera position.

$$\det(\boldsymbol{Q}\boldsymbol{R}\boldsymbol{Q}^{-1}) = \det(\boldsymbol{Q}) \det(\boldsymbol{R}) \det(\boldsymbol{Q})^{-1} = 1 \,, \tag{23}$$

it is because $\det(\boldsymbol{Q}) \neq 0$, $\det(\boldsymbol{R}) = 1$ and $\det(\boldsymbol{Q}^{-1}) = \det(\boldsymbol{Q})^{-1}$. Thus, the transformation Jacobian of the 3D points is non-zero. For the Jacobian of the transformation from $(x, y)$ to $(\bar{x}, \bar{y})$, we only need to consider the upper-left $2 \times 2$ submatrix of $\boldsymbol{Q}\boldsymbol{R}\boldsymbol{Q}^{-1}$. Since the upper-left $2 \times 2$ matrix of $\boldsymbol{R}$ represents the rotation matrix in the $xy$ plane, thus it is non-zero. The upper-left $2 \times 2$ matrix of $\boldsymbol{Q}$ is diagonal also non-zero. Hence, we get the determinant of Jacobian from $(x, y)$ to $(\bar{x}, \bar{y})$ is also non-zero. $\square$

### A.2.4 VOXEL BASED INTERPOLATING SAMPLING DETAIL

Each property of the voxel grid is stored in a vector. Hence, to get an attribute of one voxel, we need the index of the voxel. Given a random point $\mathbf{p} \in \mathbb{R}^3$, the voxel index $(i, j, k)$ for voxel $\mathbf{v}$ which contain points $\mathbf{p}$ can be localized by

$$(i, j, k) = \text{round}(\frac{1}{v_s}(\mathbf{p} - \mathbf{c})) \tag{24}$$

where $\mathbf{c} \in \mathbb{R}^3$ is the center coordinates of voxel grid. Thus, we can localize the voxel in one step without any nearest neighbor search.

### A.3 MORE VISUALIZATION RESULTS

### A.4 FAILING CASES ANALYSIS

Chibane et al. (2020b) shows successful results in ShapeNet Chang et al. (2015). However, we did not get satisfactory results on both object and scene datasets. We suspect the method needs a lot of training data, e.g., ShapeNet has multiple point clouds for a single shape. We only have one (sparse) point cloud for a shape. We also see a similar issue reported in the authors' Github issues. Fig. 14 shows different failing cases. The first line is the failing case on open surface reconstruction for sparse input ($\sim 6k$) *lr_kt0* datasets. IF-NET Chibane et al. (2020a) can not handle open surfaces. Neural-Pull Ma et al. (2020) easily fails when there is a flat plane. SIREN Sitzmann et al. (2020) tends to create artifices in non-surface areas when dealing with open surfaces. The second row is failing cases for complicated shape *Dragon* (sparse input, $\sim 6k$ points). We also fail to recover satisfactory results using our method (last column), and the modified Neural-Pull method with our sampling strategy also fails to recover the correct shape of the Dragon (last second column). Our analysis is that large details are gathered around the head part of the Dragon, and the interpolation fails to overcome too sparse input.

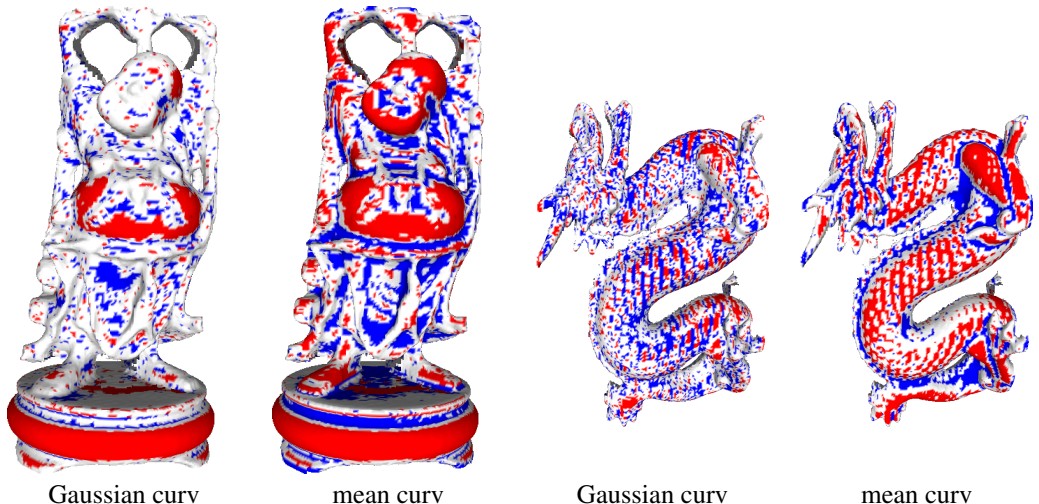

Gaussian curv       mean curv       Gaussian curv       mean curv

Figure 11: The visualization of Gaussian curvatures and mean curvatures of *happy_buddha* and *dragon* dataset, computed using depth images as described in main paper section 3.1.

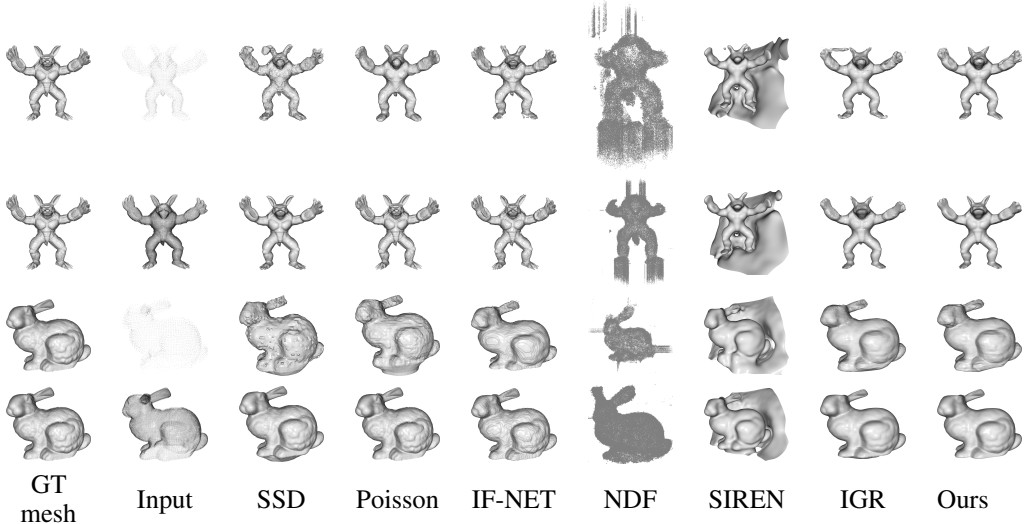

GT mesh    Input    SSD    Poisson    IF-NET    NDF    SIREN    IGR    Ours

Figure 12: Comparison results with SSD Calakli & Taubin (2011), Poisson surface recontruction Kazhdan et al. (2006), NDF Chibane et al. (2020b), IGR Gropp et al. (2020), SIREN Sitzmann et al. (2020) and IF-NET Chibane et al. (2020a) with two different density input on synthetic datasets Curless & Levoy (1996). *Armadillo* has $\sim 8k$ in sparse input and $\sim 146k$ in dense input. *Bunny* has $\sim 5k$ sparse points and dense one with $\sim 100k$ points

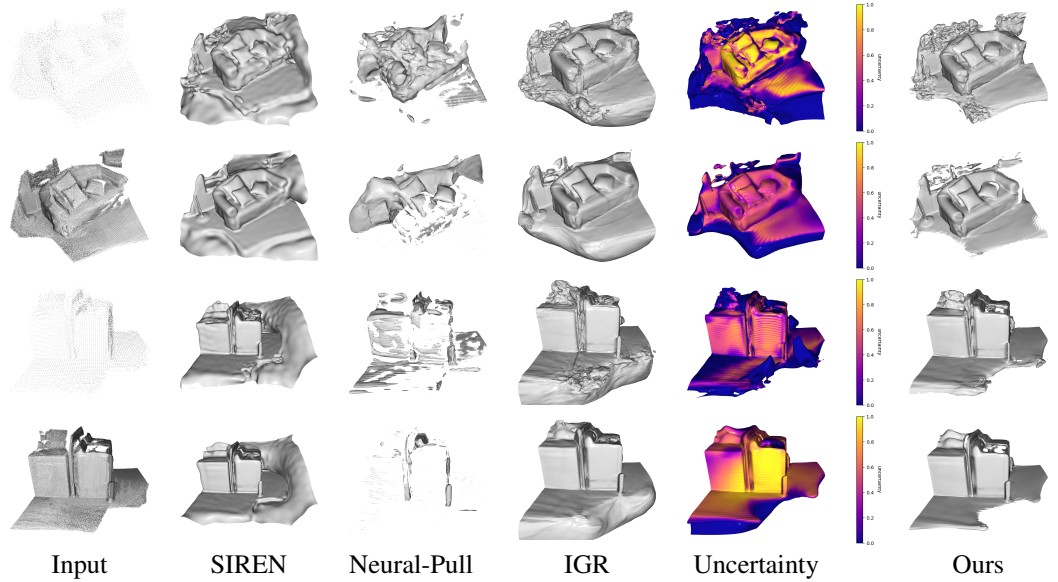

| Input | SIREN | Neural-Pull | IGR | Uncertainty | Ours |

Figure 13: Two **Real world** Scene datasets Choi et al. (2016) results comparison with SIREN Sitzmann et al. (2020), Neural-Pull Ma et al. (2020) and IGR Gropp et al. (2020). Neural-Pull fails in most of the cases, while SIREN creates a redundant area. *Sofa* has $\sim 6k$ points and $\sim 111k$ points in sparse and dense situation, respectively. *Washmachine* with $\sim 8k$ sparse points and $\sim 180k$ dense points.

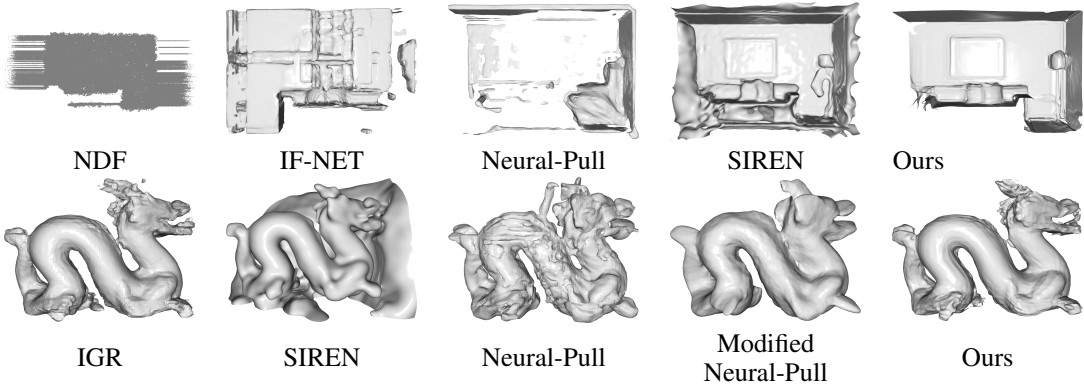

| NDF | IF-NET | Neural-Pull | SIREN | Ours |
| IGR | SIREN | Neural-Pull | Modified Neural-Pull | Ours |

Figure 14: Failing cases in different methods.

