# OpenReview forum: "NeuroSURF: Neural Uncertainty-aware Robust Surface Reconstruction"
_ICLR.cc/2024/Conference — ICLR 2024 Conference Withdrawn Submission_

### Official Review · Reviewer_MUY9 · 2023-10-22

**Soundness:** 3 good
**Presentation:** 1 poor
**Contribution:** 2 fair
**Rating:** 3
**Confidence:** 4

**Summary:**

NeuroSURF is a new method that takes posed depth images of an arbitrary static object as input and returns its SDF surface reconstruction. First, it computes mean and Gaussian curvatures of the backprojected depth in image space for each depth image. Then, the backprojected depth images/camera-space SDFs are merged in a coarse voxel grid. In addition, during merging, the curvatures are also merged and, as in a prior work on which the submission builds, each voxel also stores an uncertainty value and the gradient of the SDF. This voxel grid can then be very quickly queried to determine the curvature for arbitrary points via nearest-neighbor interpolation and uncertainty computation, which in turn allows to determine low-/middle-/high-curvature samples. Finally, these samples (positions, interpolated uncertainty, SDF gradient) are used to directly supervise a coordinate-based MLP that regresses SDF and uncertainty. I do not understand what happens afterwards. The main contribution/novelty lies in using curvature to guide the sampling and in using the uncertainty to mask out areas without supporting depth-image evidence during mesh extraction with Marching Cubes.

**Strengths:**

- The uncertainty value can be used as a mask when extracting a mesh from the SDF field via Marching Cubes. This enables open surfaces.

- The scheme can be swapped into other methods, for example IGR, to improve their results. The experiments support this claim quantitatively. The authors promise to release code.

- The results on sparse (only a few thousand points, fairly uniformly distributed) are qualitatively and quantitatively much better than prior work.

**Weaknesses:**

*Method*

- Are depth discontinuities between neighboring pixels (e.g. from occlusions) taken into account when computing depth derivatives in image space? If not, that should be stated in the limitations.

- What is the point of transferring the information of the voxel grid into an MLP? The values extracted from the voxel grid are used directly for supervision of the MLP, such that the MLP will learn to simply reproduce the extracted values. Why not then directly treat the extracted values as the surface reconstruction functions? What does storing them in an MLP do? Memory savings? But that comes at the cost of speed, no?

- An overview figure would help with putting together the components. What is happening in Sec. 3.4? I think this part would benefit from feedback by someone not closely involved in the submission. It's currently plain confusing. Does SIREN/Neural-Pull become a part of NeuroSURF? If so, what does that mean? Or is the NeuroSURF sampling used in SIREN/Neural-Pull? I.e. instead of projecting a random point onto the closest surface by using the SDF and its gradient (obtained via back-prop through the SDF MLP of SIREN/Neural-Pull), the coarse voxel grid which stores SDF values and SDF gradients is queried? Doesn't turning that voxel grid into an MLP (see previous point) defeat the point then, which presumably is speed-up?

- Are Sec. 3.1 and 3.2 the core of the method, namely sampling; Sec. 3.3 is a straightforward application to surface reconstruction; and Sec. 3.4 is *another* application of Sec. 3.1 and 3.2 to surface reconstruction, by incorporating the sampling into prior reconstruction methods? Please add introductory sentences for context.

- Please explain better which parts of the method are responsible for which important downstream advantage. This motivation is missing. Why were these design choices made? What is the goal of each step of the method section?

*Results*

- There are a number of papers that I would like the authors to comment on in the rebuttal and in a revision of the submission, especially as to why comparisons to them are not required:
* Duan et al. Curriculum DeepSDF discusses sampling/weighting schemes for training DeepSDF-like networks.
* Hanocka et al. Point2Mesh is classical in spirit and allows for adding finer resolutions without re-running the entire coarser pipeline and it shows results on quite noisy/sparse point clouds, i.e. it does not share the limitations argued in Related Work regarding classical surface reconstruction methods.
* Atzmon et al. SAL: Sign Agnostic Learning of Shapes from Raw Data is a deep implicit surface reconstruction method that works from unoriented and noisy raw point clouds, i.e. it does not share the limitations of learning-based methods mentioned in Related Work.
* Takikawa et al. Neural Geometric Level of Detail is a major implicit surface reconstruction method.
* Lindell et al. BACON: Band-limited Coordinate Networks for Multiscale Scene Representation is another major implicit surface reconstruction method.
* And a couple of papers from 2022 by Zhizhong Han beyond the 2020 paper Neural-Pull (which the submission compares to), namely Li et al. Learning Deep Implicit Functions for 3D Shapes with Dynamic Code Clouds and Ma et al. Reconstructing Surfaces for Sparse Point Clouds with On-Surface Priors.
* Also, why is there no comparison/ablation to Gradient-SDF, which appears to be the basis for the submission?

- Do SIREN, IGR and Ours use the same resolution and bounding box for Marching Cubes? IGR shows clear Marching Cubes artifacts, while the other two don't. That is relevant because IGR looks very similar to Ours except for these artifacts.

- As discussed in *Method* above, I'd like to see results when extracting values from the voxel grid directly (in the same manner as when generating samples for MLP supervision) instead of transferring them into an MLP. How does the quality change? What about memory and speed?

- In the middle of the first paragraph of Sec. 4, it says that for methods from prior work that take point clouds as input, these point clouds are obtained via the voxel grid. Does that mean that the voxel grid from the submission is used for that? Wouldn't that imply that the voxel grid, i.e. the proposed method, upper-bounds all other methods because the other ones only get as input what the submission produces as intermediate output? That seems like an unfair setup?

- There are no qualitative results that allow to assess the qualitative difference that swapping the proposed method into IGR makes.

**Questions:**

As of now, the writing is too confusing and unclear. The motivation of design choices, context, and even the overall goal of the method is unclear to me. Furthermore, I'd like to see comparisons (or a discussion as to why that's unnecessary) to works from after 2020. These two aspects are my main objections to accepting the paper.


*Minor notes*

Beyond the questions in Weaknesses, please address the following questions in a rebuttal:

- What is l_E, the last loss term in equation 9?

- Sec. 3.3 early on mentions an uncertainty threshold tau. What is it set to? 0?

- Is uncertainty ever used for anything other than as a binary mask during Marching Cubes where it is presumably "threshold-ed" at 0, i.e. parts that where never observed in the input depth images are removed?

- What is the point of the last paragraph of Sec. 3.4? Doesn't the ability to project points to the surface come from the prior work of Gradient-SDF? If so, it should be clearer that this paragraph isn't a contribution by the submission.

- Are arbitrary non-intersecting open surfaces possible? Using a mask in Marching Cubes seems like it might be restrictive? Beyond issues that would arise due to an insufficient resolution of the voxel grid?

- Are the three groups of curvature samples (end of Sec. 3.2) ever used for anything? Where in the method does it matter that they were split into three groups?

- Is there a reason why Gaussian curvature is never used in the experiments? A comparison between Gaussian and mean curvatures would be interesting for people who want to use the method. (This is optional since Gaussian curvature could also just be removed from the method section.)


These are just some other notes that do not need to be addressed:

- Not strictly necessary, but a point towards sophisticated classical shape representations would be helpful, e.g. Ohtake et al. Multi-level partition of unity implicits from 2005.

- I'm not following how Sec. 3.1 and Sec. A.2.3 are connected. Does invariance to parametrization changes (main text) relate to the Jacobian of parametrization changes having non-zero determinant (appendix)? Please make the connection to the main text more explicit in the appendix.

- Should Equation 5 use psi_p instead of psi_x? psi_x is not defined as far as I can see.

- At the end of Sec. 3.2, there is a reference to Sec. 3.2.

- I assume Figures 11 and 12 belong to Sec. A.3? A sentence in Sec. A.3 referring to them would clarify that.

- Figure 9 does not look that convincing. The bunny evolves nicely during training both with and without the proposed sampling. Another shape might demonstrate the advantages better, like the right one.

- The caption of Figure 8 should state what solid and dashed means, not just the main text on the previous page.

- Figure 5 could be easily extended by a neat ablation: a qualitative result from the full proposed method with and without the Marching Cubes thresholding.

---

> ### Author Response · Authors · 2023-11-13
> **Address method weakness**
>
> Thank you for the insightful comments and suggestions provided by the reviewers, which we believe have significantly contributed to improving the quality and clarity of our manuscript. In the following sections, we systematically address each reviewer's questions and concerns, providing clarifications and additional information where necessary.
> * **Weakness 1 (Method)**:
>     + It is true that noisy depth and missing data on depth would damage the quality of computing gradients on depth. Thus we first applied a median filter on depth before computing. For normal computing, the FASL method considers patches with window size $k$ ($k=5$ in our case), not only four neighbors. For curvatures and normals per voxel, the weighted average in equations (19) and (20) can also help reduce the noise since if depth is missing, the uncertainty value $w_i^v$ will also be small. These will cause empty areas when we generate mesh, considering uncertainty (since the uncertainty will be small/zero), which we list in the limitations that our work can not do shape completions.
>     + We assume the reviewer means that if we can build voxel ready, which can extract mesh from it, why we still need to use a neural network to learn it. The answer is that the neural representation of SDF and voxel representation fundamentally differ in both theoretical and application aspects. Theoretically, the neural network is a continuous representation. Advantages, for instance, when it comes to differentiable analysis, such as taking derivatives, the discontinuity is no longer an obstacle. For applications, when you can encode the shape into a neural network rather than in a discrete manner, it would be suitable for some downstream tasks such as dynamic object modeling. Imaging each time step you need a different voxel stored in discrete cases, for a neural network, you just need to give input $(x,y,z,t)$ to the network. That would be our future work. Let lone neural implicits representation allows easy change of the reconstructed mesh resolution as we pointed out in the paper.
>     + We can certainly add a pipeline figure. We will update our PDF. For section 3.4, since we proposed a sampling method, our work can be used to improve the neural-pull/SIREN without changing the fundamental input, architectures, and loss function. It works as follows: given a set of depths, initialize the voxel grid, and extract the point cloud directly from the voxel grid (this part is to solve non-available point cloud problem in real applications). If the point cloud is too sparse, a method like neural-pull will fail. Since neural-pull works as given point cloud $\{x_i\}_i$, it samples around the input points $\{\hat{x}_i\}_i$, then it runs the nearest neighbor search to pair $\hat{x}_i$ with a point in the point cloud $x_i$, then the network learns to pull $\hat{x}_i$ to $x_i$. But if the input point cloud is too sparse, the nearest neighbor search fails. That is why neural-pull results in Fig 7 are bad. Using equation (4), one can skip the nearest neighbor search by directly localizing the surface point $x_i$ using equations (1) and (4) for $v_i$ and $v_p$. And learn to pull $v_i$ and $v_p$ to $x_i$.
>     + Yes. We will modify our manuscript for a clearer expression.
>     + We have 3 core part: 1) sec 3.1 integrate curvature into voxel which enables curvature-guided sampling. 2) sec 3.2 interpolation sampling deals with sparse extracted point clouds. 3) 3.3 Uncertainty estimation which enables open surface reconstruction. The overall motivation: 1) chose voxels: as appreciated by reviewer PKLg, depth images can be cheaply available from the depth sensor. The following question will be, why not use depth directly? One can get the point cloud by re-projecting depth using camera poses. The answer is noise and camera poses. Since we will need the camera poses anyway, we can do it jointly when integrating depth into a voxel structure. In the experiment shown in Figure 10, real-world dataset (TUM\_rgbd) and Figure 13 (appendix) dataset red\_wood, we all estimated camera poses when we computed normals and curvatures. The second reason is noise. As explained in the first point. 2) curvature-guided sampling: The first reason, as demonstrated in Figure 8 and Figure 9 and explained in section 2 (Related works, biased sampling), is that curvature-guided sampling enables faster convergence. Second reason: We extract surface points using equation (1), which is already studied in Yang et al. 2021 (Geometry processing with neural fields), leading to a concentration on high curvature areas. Curvature-guided sampling solves these biases. 3) interpolation sampling: deals with sparse situations and bypasses the nearest neighbor search.

---

> ### Author Response · Authors · 2023-11-13
> **address weakness in results and questions**
>
> * **Weakness (results)**:
>     + We thank the reviewer for pointing out all related work. We'll add it to our citations. But the main reason we did not compare against all relevant methods is we focused on improving sampling, enabling curvature information, and using depth as input. That is why we only choose the most representative methods to show our ability to improve it. Moreover, as we mentioned, the final goal is beyond just reconstructing a mesh from the point cloud. We would like to have neural implicits representation which might potentially benefit for more downstream tasks. A deeper reason is that we think the trained neural implicit representation can enable more applications such as the deformation of the shape, which we plan to do next. For example, the dynamic object would be much easier recorded by a RGB-D camera.
>
>     + marching cube resolution: all results are rendered using 128 voxel resolutions.
>     + We can certainly add them.
>     + on the contrary. The reason we give comparison methods the point cloud extract from the voxel grid is because these are more clear and oriented point clouds. Considering the original input, which are depth maps, we could give them the point cloud generated by re-projecting depth into 3D world. Then honestly we will overperformance all comparison methods a lot since the point cloud is noisy (especially in real-world dataset) and doesn't have normals.
>     + We can add it of course. We will add it in the appendix.
>
> * **Questions**:
>     + $l_e$ is an Eikonal loss, which ensures the optimized neural network indeed is an SDF field. (we will update the PDF).
>     + it is a hyperparameter, we set to $\tau=1e-3$ during mesh extraction.
>     + yes. the point has zero uncertainty due to missing data on depth, which could be because of open areas or occlusions. For occlusion is the limitation of our method which we listed in the limitation section.
>     + Theoretically, yes. As the neural implicits should represent $f(x, \theta)$ the continuous SDF, which shouldn't have a problem for representing intersections. In practice, depending on sampling and training steps, even the marching cube step could have failure cases on heavily self-intersected areas. The interpolation is designed to complement the inefficient resolution of the initialized voxel grid.
>     + Please refer to the biased sampling part in related work and also the answers for the method weakness part.
>     + main curvature $H = k_1 + k_2$, Gaussian curvature is $K = k_1k_2$, It is certainly true that main curvature and Gaussian curvature capture different geometry information. Since we are not using curvature values directly, rather than using them as an index to indicate the geometry feature of the surface, and if the reviewer see Fig 2 and Fig 11, the distribution of main and Gaussian curvature are similar. That means choosing a slightly different threshold to divide the low, median, and high curvature categories when using two different curvatures should have relatively similar sampling results. We will add an ablation example to demonstrate and also add an explanation in the appendix.  A more mathematical way of think about it is $G = K_1k_2$, $H^2 \geq 4G$, $H$ is kind of a upper bound of $G$ since $|H| \geq 2\sqrt{|G|} $. So, there is no specific reason that we chose the main curvature rather than the Gaussian curvature.

---

> > ### Comment · Reviewer_MUY9 · 2023-11-14
> > **Thank you for the response**
> >
> > Thank you for the response!
> >
> > The additional explanations about the writing/presentation are convincing to me.
> >
> > Regarding the experiments, I still have questions:
> >
> > - If a neural implicit is the goal, rather than a mesh, then Curriculum DeepSDF, SAL, NGLOD, and BACON are still viable methods. Especially SAL deals with noisy input, so it tackles the same problem as the submission. As of now, the submission demonstrates that its sampling scheme can be plugged into older methods like IGR or Neural Pull to improve them. But what if newer methods already give better results? Would the new sampling scheme still contribute? If not, then why should Neural Pull+submission be used instead of newer methods? I don't really understand the argument from the rebuttal as to why such comparisons are not necessary.
> >
> > - Is the bounding box used by Marching Cubes the same for all methods? If the MC resolution is the same, then a difference in bounding boxes is the only explanation I have for IGR's artifacts. Is there another explanation for these artifacts? If not, then this is important because the current qualitative results allow for the possilibity that IGR with proper MC settings might be on par with the submission.
> >
> > - I see. Please also state directly in the method section how the low, middle, and high curvature points are used.

---

> > > ### Author Response · Authors · 2023-11-17
> > >
> > > We want to thank all reviewers for their constructive opinions. Unfortunately, we cannot address all the experiments due to the time conflicts with the CVPR deadline. Thus, we decided to withdraw the paper. It is a pity that we met all good reviewers but couldn't update our work.

---

### Official Review · Reviewer_PKLg · 2023-10-29

**Soundness:** 2 fair
**Presentation:** 2 fair
**Contribution:** 2 fair
**Rating:** 3
**Confidence:** 3

**Summary:**

The manuscript introduces a pipeline designed for the reconstruction of surfaces from depth images. NeuroSURF is founded upon the voxelized SDF representation initially proposed by [Sommer et al, 2022]. It leverages a mean-curvature guided sampling approach, coupled with uncertainty values, to facilitate the extraction of open surfaces. NeuroSURF's performance was assessed across diverse datasets, encompassing synthetic scenes featuring ideal depth images, as well as real-world scans of indoor environments and objects.

**Strengths:**

-	The problem is of interest to the research community, as depth images can be cheaply available from depth sensors.
-	The formulation of open-surface extraction is intuitively sound, which has been a common issue for SDF representations.
-	The experiment setup is diverse, including real-world and synthetic datasets. This setup helps readers understand the values of the proposed approach.

**Weaknesses:**

- Theoretical Limitations:

  - **Curvature Sampling Limitations**: The utilization of curvature information to capture high-frequency surface details introduces certain limitations when employing mean and Gaussian curvatures. Sampling-based on mean curvature may overlook saddle regions where the curvature is positive in one direction but negative in another. Similarly, sampling based on Gaussian curvature might miss developable regions where one direction exhibits zero curvature while the other has high curvature. Although Figure 11 demonstrates the distinct biases of Gaussian and mean curvatures in sampling, the manuscript does not address their potential impact on performance.

  -  **Noise Sensitivity in Curvature-Based Sampling**: Sampling-based on curvature inherently exhibits sensitivity to noise. In the presence of a noisy depth map, all pixels tend to exhibit high curvature and, consequently, receive heavy sampling. The manuscript does not discuss strategies to mitigate this sensitivity to data noise effectively.

  - **Discontinuities in Voxelized Representation**:  The adoption of a voxelized representation with only a single center point and no interpolation between neighboring voxels can inherently introduce discontinuities in the SDF, as inferred from Equation 4 and the descriptions on Page 5.



- Inadequate Experiment Results:
  - While NeuroSURF is compared against many recent generic representations such as SIREN and application-specific approaches such as Neural-Pull, the baselines in both cases are somewhat inadequate.
    - More recent generic representations such as Instant NGP [Müller et al, 2022] are not included, which potentially can resolve the lack of high frequency details issue.
    - As for converting depth/point cloud to surfaces, there are also techniques such as recent NKSR [Huang et al, 2023], and differentiable possion SAP [Peng et al, 2021], which are robust to noise while recovering the geometric details.
    - The formulation (Equation 18-21) proposed in Appendix A.2.2 is similar to truncated signed distance function (TSDF) [Curless and Levoy, 1996]. In this case, I wonder if TSDF is adequate enough for recovering the surfaces accurately.

-	Unclear descriptions:
  - The manuscript lacks a clear explanation of how uncertainties and SDF values are computed from the depth images. This information is crucial for understanding the constraints and potential limitations of the surface recovery process from depth data.
  - The use of notations within the manuscript can be confusing. It remains unclear what represents network estimates and what are the inputs, despite the presence of Table 4 in the appendix. Additionally, the manuscript does not provide formal definitions for certain notations like $\psi^x$ in Equation 5.


References:

Müller, T., Evans, A., Schied, C., & Keller, A. (2022). Instant neural graphics primitives with a multiresolution hash encoding. ACM Transactions on Graphics (ToG), 41(4), 1-15.

Huang, J., Gojcic, Z., Atzmon, M., Litany, O., Fidler, S., & Williams, F. (2023). Neural Kernel Surface Reconstruction. In Proceedings of the IEEE/CVF Conference on Computer Vision and Pattern Recognition (pp. 4369-4379).

Curless, B., & Levoy, M. (1996, August). A volumetric method for building complex models from range images. In Proceedings of the 23rd annual conference on Computer graphics and interactive techniques (pp. 303-312).

Peng, S., Jiang, C., Liao, Y., Niemeyer, M., Pollefeys, M., & Geiger, A. (2021). Shape as points: A differentiable poisson solver. Advances in Neural Information Processing Systems, 34, 13032-13044.

**Questions:**

-	Given the limitations of mean/gaussian curvatures, could there be some ablations to further understand the impact on performance? Can the manuscript use total curvature instead?
-	How sensitive the NeuroSURF is w.r.t noise? Could the manuscript provide some ablations?
-	Can the authors provide clarifications on if there are discontinuities across voxels?
-	Can the authors provide justifications for why some of the sensible baselines are not included? If they are indeed sensible baselines, could the manuscript include their results?
-	Could the authors provide clarifications on the preprocessing of depth images to obtain SDF and uncertainties?
-	Could the manuscript clarify what are the optimizable variables and what are the inputs?

---

> ### Author Response · Authors · 2023-11-13
> **Address Weakness**
>
> Thank you for the insightful comments and suggestions provided by the reviewers, which we believe have significantly contributed to improving the quality and clarity of our manuscript. In the following sections, we systematically address each reviewer's questions and concerns, providing clarifications and additional information where necessary.
> * **weakness 1 (Theoretical limitations)**:
>     + Curvature sampling limitation: we assume reviewer means for main curvature $H = k_1 + k_2$, and saddle region $k_1 > 0$, $k_2$ negative. This will cause $H$ to be relatively small. Since we divide $H$ into low, median, and high categories, this situation will be classified into the low $H$ category. It is certainly true that main curvature and Gaussian curvature capture different geometry information. Since we are not using curvature values directly, rather than using them as an index to indicate the geometry feature of the surface, and if the reviewer see Fig 2 and Fig 11, the distribution of main and Gaussian curvature are similar. That means choosing a slightly different threshold to divide the low, median, and high curvature categories when using two different curvatures should have relatively similar sampling results. We will add an ablation example to demonstrate and also add an explanation in the appendix.  A more mathematical way of think about it is $G = K_1k_2$, $H^2 \geq 4G$, $H$ is kind of a upper bound of $G$ since $|H| \geq 2\sqrt{|G|} $.
>     + noise sensitive: that is one reason we chose voxel as the base structure. Indeed, the curvature is computed per pixel using its neighborhood pixel depth. However, we did not choose to directly use point cloud generated by re-projecting depth to 3D, but rather fuse them into a voxel grid, other than enable easier localization when sampling. Another reason is it reduces the noise by taking a weighted average of both gradient, curvature, and SDF. See equation (18-21). Another step we did to reduce the noise is a median filter is applied for each depth before computing normals and curvatures. We will add it to the paper.
>     + Discontinuities in voxel: that is certainly a disadvantage of classical surface representation and motivates us to continue to train a neural network after initializing the voxel. The key point is, that even though it is a discrete representation, we could have a continuous approximation on each point near the voxel center. But the reviewer is totally correct that if the voxel grid is too coarse, there will be more discontinuity on the original grid. That is why we did 64 and 256 voxel resolution experiments to show that under the coarse, discrete case, our sampling can reduce the discontinuity by Taylor expansion (equation 4) to improve the results.
> * **Inadequate Experiment Results**:
>     + The reason why we compare to baselines such as IGR and neural-pull is they are the most fundamental methods and our contribution is focused on improving sampling, bringing a bridge between more real applications (use depth, instead of already existing point cloud), and dealing with unbiased sampling situations. And easily swap to other methods as reviewer MUY9 appreciated. But we thank the reviewer for pointing out the related works, we will certainly cite them and for the new paper Neural Kernel Surface Reconstruction we can try to add a comparison against it.
>     + instant NGP: We also thought about after instantNGP came out, should we all go for neural rendering? But we still believe incorporating depth information would benefit in different ways. As we focus on improving the sampling and show more possibility of using more information from depth images. A deeper reason is that we think the trained neural implicit representation can enable more applications such as the deformation of the shape, which we plan to do next. For example, the dynamic object would be much easier recorded by a RGB-D cameras.
>     + TSDF: indeed we used TSDF, as TSDF means truncated SDF, which is a way to truncate SDF value when updating the voxel distance, but the way we compute distance $d_s$ is different as the pre-computed gradient allows us to project voxel to surface plane rather than compute point-to-point distance. The gradient voxel structure is based one paper gradient-SDF (sommer et al). on 2022 with modified curvature attributes stored as well. According to the paper, the voxel integration performance is comparable with other state-of-the-art methods.  More importantly, the new voxel data structure allows us to do Taylor expansion (equation 4) without the nearest neighbor search.

---

> > ### Author Response · Authors · 2023-11-13
> > **Address Questions**
> >
> > * **Unclear descriptions**:
> >     + uncertainty and SDF: We are very sorry that due to the page limitation, we moved this part totally to the appendix, we kindly ask the reviewer to refer to section A.2.2 in the appendix for more details. We briefly summarize the process here: Given a depth image $D_i$, we use FASL (Fast and accurate computation of surface normals from range images) and equation (3) to compute normal and curvatures per points, then use equation (12-17) to integrate voxel normal/curvature/uncertainty to each voxel. When the camera poses are missing, we jointly estimate the camera pose during these steps (we can add more details if the reviewer requires the camera pose part). uncertainty computation in voxel initialization (section 3.1) used equations (14-16) in the appendix (we put them in the appendix because it is a standard procedure for classical voxel-based method). Initiatively, given an already existing point cloud (let's say we already integrated the first $k-1$ depth, now we deal with depth $k$), after transforming points from depth $k$ to world coordinates (equation 12), we now compare current voxel to the incoming points from depth $k$, as we assume points from depth $k$ are on the surface of the object (should have $0$ SDF value), then voxel center $v_i$ are projected to surface plane to compute the distance (equation 14) with the help of normal $g_i$, the uncertainty (weight) $w_i$ is determined by the value of SDF $d_s(v_i)$. Then, the positive $d_s$ means this voxel is seen by this depth without occlusion or out of the visible area. The linear part in equation 16 is a tolerance of certain errors (up to $v_sT$), the 0 $w_i$ means this voxel is not visible/does not have a corresponding point to update the distance value $d_s$. Thus, after integrating all depths, a voxel with high $w_i$ has been seen (updated) by more depth images with smaller errors, thus the distance value is more reliable. $0$ uncertainty parts indicated empty area or occluded part. We will certainly update our manuscript to provide more details.
> >     + Notation: Sorry for the mistake, $\psi^x$ in equation (5) should be $\phi_p$, we will correct it. The neural network input and output are described in the beginning of section 3. We will update and emphasize in new manuscript.
> > * **Questions**:
> >     + ablation for Gaussian/main curvature: We will add a sampling ablation to show the sampling results of main curvature-guided and Gaussian curvature-guided results.
> >     + We could certainly do an ablation on noise.
> >     + The reason we did not include SAP (Peng, S.,2021) is we focus on improving the sampling part and bridging the gap between real applications using depth images, which allows us to get normals/curvature information. Thus, we build on some fundamental points to implicits methods. The difference for us with SAP is that the goal of SAP for a learning-based setting is to predict a clean-oriented point cloud and then use Poisson solver to reconstruct the mesh. But first, our fusion to voxel can compute normal already and the weighted average in equation (18) eliminated most of the noise already. Plus, this paper needs "watertight and noise-free meshes for supervision" which is against our goal of real application, easily acquired input, and no ground truth.
> >     + We will certainly update the manuscript for a better description of depth processing and optimization variables. Please give us ~2 days.

---

> > > ### Author Response · Authors · 2023-11-17
> > > **Thank you**
> > >
> > > We want to thank all reviewers for their constructive opinions. Unfortunately, we cannot address all the experiments due to the time conflicts with the CVPR deadline. Thus, we decided to withdraw the paper. It is a pity that we met all good reviewers but couldn't update our work.

---

### Official Review · Reviewer_Mh1P · 2023-10-30

**Soundness:** 3 good
**Presentation:** 2 fair
**Contribution:** 2 fair
**Rating:** 5
**Confidence:** 4

**Summary:**

The paper introduces a technique for surface reconstruction from depth maps using an optimization-based approach. Central to this method is the emphasis on proficient sampling and filtering of input points, which are then integrated into a reconstruction method that employs an implicit field as its underlying representation. The filtering process proceeds by transforming the initial point cloud into voxels and adeptly determining attributes for these voxels, such as the SDF, gradient vector, and curvatures. Consequently, the optimization-based methods receive sampled points derived from the voxel grid structure. Beyond just surface attributes, the implicit field can incorporate surface uncertainty, facilitating intuitive open surface extraction and noise mitigation. When compared with several prevailing methods like Poisson Surface Reconstruction and IGR, the proposed sampling method's efficacy becomes evident.

**Strengths:**

- The technique addresses the innovative challenge of point cloud sampling specifically tailored for neural reconstruction from depth maps. It harnesses the voxel grid structure as a foundation to accelerate both computation and sampling processes.

- A distinctive feature of the method is its ability to produce uncertainty as an auxiliary output. This capability paves the way for a bunch of subsequent applications, including navigation.

- The experiments conducted are exhaustive, and the ablation analysis adeptly showcases the potency of each individual component.

**Weaknesses:**

- The motivation is somewhat unconvincing. The justification for adopting biased sampling remains ambiguous, and the choice to utilize voxel-based sub-samples over complete samples isn't adequately described.

- The section detailing the method is ambiguous, with many important details being omitted. This absence hinders the algorithm's reproducibility. Please see the 'Questions' section for a more in-depth breakdown.

- The computation of the uncertainty lacks a straightforward explanation. Specifically, where does the uncertainty come from? Are they coming from the quantization artifact introduced by voxelizing the points or coming from the sensor themselves (for example the angle between the surface normal and camera ray)? The authors should provide some intuitive examples demonstrating low and high confidence areas.

- The method's innovation is questionable. Employing curvature as a guiding principle for sampling isn't a novel approach (as already pointed out in the related works section). Moreover, the inherent characteristics of depth maps don't appear to be optimally leveraged.

**Questions:**

- How is the voxel grid built? How are the attributes such as gradient initialized for each voxel grid?

- Why is the geometry within each voxel being approximated as planar patches? Could fitting primitives such as ellipsoid or parabolic surfaces improve the results?

- Why is the uncertainty smaller if the points are far away from the voxel, as explained on Page 4?

- Fig.4 is not clear. Why would the sample still be evenly distributed on the surface given the curvature-aware sampling?

**Details Of Ethics Concerns:**

Not applicable.

---

> ### Author Response · Authors · 2023-11-13
> **Address weakness part**
>
> We would like to extend our gratitude for your valuable feedback and expert insights on our paper. We are especially grateful for your positive feedback on the technical aspects of our work. We have carefully reviewed each of your points and are pleased to answer your comments and questions.
> * **weakness 1 (motivation)**: Here we briefly explain our motivation for choosing voxel as a base structure and curvature-guided sampling and we will explain technical details in **Question section**.
>     + Voxel structure: as appreciated by reviewer PKLg, depth images can be cheaply available from the depth sensor. The following question will be, why not use depth directly? One can get the point cloud by re-projecting depth using camera poses. The answer is noise and camera poses. Since we will need the camera poses anyway, we can do it jointly when integrating depth into a voxel structure. In the experiment shown in Figure 10, real-world dataset (TUM\_rgbd) and Figure 13 (appendix) dataset red\_wood, we all estimated camera poses along when we computed normals and curvatures. The second reason is noise. In real applications, acquired depth can be very noisy. Directly re-project them to 3D space will end up in a very noisy point cloud. Even though each point still has computed normal and curvature, these attributes will be less accurate as we fuse it to a voxel and then sample from it. The computation of normal and curvature depends on the quality of depth. Unless one extends the pipeline to do a depth pre-processing/denoise/complication first, normals and curvature only computed from a single depth won't be robust. However, please see the equations (18-21) in the appendix. One can tell that the normal and curvature integrated into voxels are weighted averages. The weight is computed from equation (16), which can potentially average out normals from large error SDF points. The third reason is uncertainty. Without voxel integration, we do not have weights as prior to train the neural network to learn the uncertainty. Please see section equation (5) on how uncertainty values are obtained.
>     + curvature sampling: The first reason, as demonstrated in Figure 8 and Figure 9 and explained in section 2 (Related works, biased sampling), curvature guided sampling enables faster convergence. Second reason: We extract surface points using equation (1), which is already studied in Yang et al. 2021 (Geometry processing with neural fields), leading to a concentration on high curvature areas. The third important fact about curvature guided sampling: sampling using **(principle)** curvature has been done in Nevello et al. 2022 (Exploring differential geometry in neural implicits), but please note that they need **ground truth mesh** to compute the curvature then use it to do curvature-guided sampling to reconstruct the mesh back again. To the best of our knowledge, we are the first method that computes and incorporates **main** curvature from depth to facilitate the reconstruction.
> * **weakness 2 (computation of uncertainty)**: uncertainty computation in voxel initialization (section 3.1) used equations (14-16) in the appendix (we put them in the appendix because it is a standard procedure for classical voxel-based method). Initiatively, given an already existing point cloud (let's say we already integrated the first $k-1$ depth, now we deal with depth $k$), after transforming points from depth $k$ to world coordinates (equation 12), we now compare current voxel to the incoming points from depth $k$, as we assume points from depth $k$ are on the surface of the object (should have $0$ SDF value), then voxel center $v_i$ are projected to surface plane to compute the distance (equation 14) with the help of normal $g_i$, the uncertainty (weight) $w_i$ is determined by the value of SDF $d_s(v_i)$. Then, the positive $d_s$ means this voxel is seen by this depth without occlusion or out of the visible area. The linear part in equation 16 is a tolerance of certain errors (up to $v_sT$), the 0 $w_i$ means this voxel is not visible/does not have a corresponding point to update the distance value $d_s$. Thus, after integrating all depths, a voxel with high $w_i$ means it has been seen (updated) by more depth images with smaller errors, thus the distance value is more reliable. $0$ uncertainty parts indicated empty area or totally occluded part. That is why we use interpolated uncertainty to train our network to indicate the uncertainty and enable open surface representation by modified Marching cubes. Regarding the occlusion or empty area caused by missing data, we listed in limitations that our work does not cover shape completions.

---

> > ### Author Response · Authors · 2023-11-13
> > **Address questions part**
> >
> > * **Question 1 (voxel initialization)**: We are very sorry that due to the page limitation, we moved this part totally to the appendix, we kindly ask the reviewer to refer to section A.2.2 in the appendix for more details. We briefly summarize the process here: Given a depth image $D_i$, we use FASL (Fast and accurate computation of surface normals from range images) and equation (3) to compute normal and curvatures per points, then use equation (12-17) to integrate voxel normal/curvature/uncertainty to each voxel. When the camera poses are missing, we jointly estimate the camera pose during these steps (we can add more details if the reviewer requires for camera pose part).
> > * **Question 2 (planar approximation)**: Regarding the planar patches approximation, we assume that the reviewer means as demonstrated in Fig 3. In principle, one could use a parabolic curve or even higher order curve to approximate the surface within one voxel, however, with what we have already, i.e. distance gradient and distance on this voxel, we can directly apply first-order Taylor expansion (equation 4), which is a linear (planar) approximation of the surface. If one would like to approximate using higher order Taylor expansion, for example, a second order for a parabola approximation, the hessian of distance needs to be computed (3x3) matrix for each voxel and not already stored as the gradient. That is why we chose first-order Taylor expansion (planar approximation).
> > * **Question 3 (uncertainty)**: Since the sampled point and its attributes are computed by Taylor expansion, a clear fact of the Taylor expansion is, that the closer to the interpolated point, the more accurate the interpolated value will be, thus the further to the voxel, the smaller uncertainty value are set (we hope it is not confused by the name, since as explained in page 3 section 3.1, larger $w_i$ indicates reliable SDF).
> > * **Question 4 (sampling distribution)**: The curvature-guided sampling idea is to prevent sampled points from being gathered on the high curvature area as the left part in Fig 4. The right part is the sampling results that are evenly sampled from 3 curvature categories, which means we consider low, median, and high areas and uniformly sampled from them. The right part shows that the results are not gathered anymore. But if it is evenly distributed on the whole surface will be influenced by the threshold setting and portion setting. If one chose to evenly divide the low, median, and high curvature area and decide the portion according to the number of points in each category, one could achieve evenly distributed sampling.

---

> > > ### Author Response · Authors · 2023-11-17
> > >
> > > We want to thank all reviewers for their constructive opinions. Unfortunately, we cannot address all the experiments due to the time conflicts with the CVPR deadline. Thus, we decided to withdraw the paper. It is a pity that we met all good reviewers but couldn't update our work.

---

### Official Review · Reviewer_drJN · 2023-11-01

**Soundness:** 2 fair
**Presentation:** 3 good
**Contribution:** 3 good
**Rating:** 5
**Confidence:** 3

**Summary:**

This paper addresses the problem of surface reconstruction from sparse input depth images/point cloud and is based on implicit neural shape representations. The key idea is a curvature guided sampling strategy that should help to improve reconstruction quality for sparse inputs by correcting for unevenly distributed points and enable interpolation among spares inputs. Furthermore the authors suggest integrating uncertainty in the loss and during the surface extraction which enables extracting open surfaces from implicit neural representations. The authors conduct experiments on object-level datasets as well as real-world datasets.The quantitative comparison on the object level shows especially good performance in the sparse inputs case.

**Strengths:**

The paper is well written and technical sound.
The curvature term is well explained and easy to follow how it is integrated in the method.
The idea of incorporating uncertainty to model open spaces with neural representations is novel and interesting.
Comparison with valid baselines is provided.

**Weaknesses:**

1) The loss function contains four parts. I’m missing an ablation study on the impact of each part for the final results, e.g. it stays unclear how much impact to the smoothness comes from the normal regularizer vs. from the interpolation sampling.
2) The paper does not discuss the network architecture of f(x, theta) in equation . As shown in previous works (SIREN, Fourier Features), there is a huge impact of the actual network architecture and input features to the final output shape, as the network acts as predefined prior. However, there is no explanation of the used architecture.

**Questions:**

It would be great to get more insights wrt. the loss function as mentioned in the weakness 1.

Regarding weakness 2) the role of f(x, theta) remains unclear to me. Please provide more explanation on that.

---

> ### Author Response · Authors · 2023-11-13
> **Ablations and Architectures**
>
> We thank the reviewer for the insightful feedback and appreciate their recognition of our technical contributions. In response to the weaknesses and queries raised, we provide the following clarifications and commitments:
> * **weakness 1 (ablation study for four loss parts)**: We can certainly add the ablation study for the four loss parts. And expected results will be
>     + 1) without normal loss $l_\mathcal{N}$ there might be challenges in accurately reconstructing complex geometries due to uncertainties in surface orientation.
>     + 2) without uncertainty loss $l_\mathcal{W}$ will cause some artifacts on open areas (similar to Figure 10 IGR/siren results).
>     + 3) without Eikonal loss $l_\mathcal{E}$ may result in trivial solutions for the optimized function $f(x,\theta)$ (e.g. solution equation 10), potentially equating to zero across most areas. This loss ensures that $f(x,\theta)$ adheres to an SDF function field.
>
> Another reason that we did not exclude the normal term $l_\mathcal{N}$ is that our pipeline allows us to compute the normals (sec 3.1) with curvature. In our situation, the normal information is always available, unlike IGR or SIREN which need to deal with no normal situation. But we can certainly add the ablations. Please give us two days. We will include it in the appendix.
> * **Weakness 2 (network architecture)**: In Section 4 (Evaluation), we mentioned our use of an 8-layer MLP with 256 nodes and ReLU activations. We want to emphasize that our focus was on sampling and interpolation, and our method adopted the IGR network architecture. This choice was made to demonstrate that our method could be seamlessly integrated with other approaches, as appreciated by reviewer MUY9. When combined with other methods, such as Neural-Pull (illustrated in Figure 7), we only modified the sampling part, keeping the Neural-Pull's network architecture unchanged.
> * **Question 1**: we will update our PDF and add the ablation study. Please give us ~2 days.
> * **Question 2**: the architecture of $f(x, \theta)$ aspects as explained before. The mathematical insights of $f(x, \theta)$ are explained in section 3 (Method). Essentially, the function $f(x, \theta)$, represented by a neural network (MLP), takes points location $x$ as input, $\theta$ is the trainable parameter. The output generated by this function is the Signed Distance Field (SDF) value of the point, along with an uncertainty value. This uncertainty metric reflects the reliability of the output SDF value.  It is crucial for $f(x, \theta)$ to adhere to the mathematical constraints of an SDF, which is the primary reason for incorporating the Eikonal loss in our framework.

---

> > ### Author Response · Authors · 2023-11-17
> >
> > We want to thank all reviewers for their constructive opinions. Unfortunately, we cannot address all the experiments due to the time conflicts with the CVPR deadline. Thus, we decided to withdraw the paper. It is a pity that we met all good reviewers but couldn't update our work.